# Filopodia powered by class x myosin promote fusion of mammalian myoblasts

David W Hammers[1,2], Cora C Hart[1,2], Michael K Matheny[1,2], Ernest G Heimsath[3], Young il Lee[1,2], John A Hammer III[4], Richard E Cheney[3], H Lee Sweeney[1,2]*

[1]Department of Pharmacology & Therapeutics, University of Florida College of Medicine, Gainesville, United States; [2]University of Florida Myology Institute, Gainesville, United States; [3]Department of Cell Biology & Physiology and Lineberger Comprehensive Cancer Center, University of North Carolina at Chapel Hill School of Medicine, Chapel Hill, United States; [4]Cell Biology and Physiology Center, National Heart, Lung and Blood Institute, Bethesda, United States

**Abstract** Skeletal muscle fibers are multinucleated cellular giants formed by the fusion of mononuclear myoblasts. Several molecules involved in myoblast fusion have been discovered, and finger-like projections coincident with myoblast fusion have also been implicated in the fusion process. The role of these cellular projections in muscle cell fusion was investigated herein. We demonstrate that these projections are filopodia generated by class X myosin (Myo10), an unconventional myosin motor protein specialized for filopodia. We further show that Myo10 is highly expressed by differentiating myoblasts, and Myo10 ablation inhibits both filopodia formation and myoblast fusion in vitro. In vivo, Myo10 labels regenerating muscle fibers associated with Duchenne muscular dystrophy and acute muscle injury. In mice, conditional loss of *Myo10* from muscle-resident stem cells, known as satellite cells, severely impairs postnatal muscle regeneration. Furthermore, the muscle fusion proteins Myomaker and Myomixer are detected in myoblast filopodia. These data demonstrate that Myo10-driven filopodia facilitate multinucleated mammalian muscle formation.

*For correspondence:
lsweeney@ufl.edu

Competing interest: The authors declare that no competing interests exist.

## Introduction

The development, growth, and repair of vertebrate skeletal muscle is largely mediated by the ability of myoblasts to fuse with each other and with pre-existing muscle fibers. In postnatal muscle, this fusion process is initiated by the activation of muscle-resident stem cells, known as satellite cells, which normally remain in a quiescent state positioned between the sarcolemma and basement membrane of muscle fibers (*Mauro, 1961*). Following an activation stimulus, satellite cells give rise to myoblast progeny which proliferate, differentiate, and fuse to bring the muscle back to homeostasis (*Charge and Rudnicki, 2004*).

The fusion of mammalian myoblasts requires the merging of two apposing lipid bilayers and has been shown to involve several widely expressed protein classes, including cytoskeleton elements (*Charrasse et al., 2006*; *Randrianarison-Huetz et al., 2018*; *Vasyutina et al., 2009*), phagocytosis receptors (*Hamoud et al., 2014*; *Hochreiter-Hufford et al., 2013*; *Park et al., 2016*), and calcium-sensing membrane repair proteins (*Leikina et al., 2013*; *Posey et al., 2011*). Myomaker (*Millay et al., 2013*) and Myomixer (aka Myomerger and Minion; the product of the *Gm7325* gene; *Bi et al., 2017*; *Quinn et al., 2017*; *Zhang et al., 2017*) have been identified as being membrane proteins essential for myoblast fusion. Current evidence suggests that Myomaker is required by both fusing cells, while the requirement for Myomixer is only unilateral (*Quinn et al., 2017*). However, the mechanisms regulating the function of these proteins are currently unknown.

Thin, actin-filled projections have been observed during the fusion of murine (*Randrianarison-Huetz et al., 2018*) and zebrafish (*Gurevich et al., 2016*) myoblasts. These structures appear to be filopodia,

which are thin, membrane-enclosed projections of actin bundles that are important for cellular behaviors such as path finding during migration, interaction with the extracellular matrix, and cell-to-cell communication (*Mattila and Lappalainen, 2008*). The actin cytoskeleton is well established as an enactor of *Drosophila* myoblast fusion (*Abmayr and Pavlath, 2012*; *Chen, 2011*), with filopodia suggested to be involved in this process (*Segal et al., 2016*). Despite actin cytoskeletal involvement in the fusion of both vertebrate and arthropod myoblasts, no members of the myosin superfamily of molecular motors have been reported to have a direct role in the muscle fusion process.

Myosin superfamily members perform actin-associated functions in all cell types. This includes the conventional (class II) myosins, such as those that power muscle contraction, and several classes of 'unconventional' myosins, of which 11 are expressed in mammals (*Berg et al., 2001*; *Odronitz and Kollmar, 2007*). These unconventional myosins employ the same basic motor mechanism as conventional myosins, but have unique tail domains that allow them to perform specialized cellular functions. Class X myosin (Myo10) is an unconventional myosin involved in the formation and elongation of filopodia

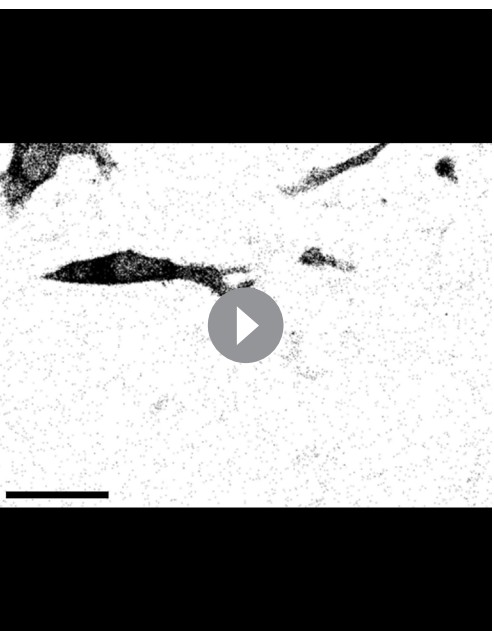

**Video 1.** Movement of undifferentiated myoblasts. Time-lapse confocal imaging of undifferentiated myoblasts expressing RFP-CAAX. Individual frames were utilized to make Figure 1A. Images were acquired every 15 min. Scale bar represents 25 μm. https://elifesciences.org/articles/72419/figures#video1

in mammalian cells (*Berg and Cheney, 2002*). Myo10 consists of an N-terminal motor domain, a lever arm with three calmodulin-binding sites and a single alpha helical domain, and a C-terminal tail containing a pleckstrin homology (PH), a myosin tail homology 4 (MyTH4), and a 4.1/Ezrin/Radixin/Moesin (FERM) domain (*Kerber and Cheney, 2011*). Upon activation, Myo10 forms an anti-parallel dimer that is optimized for the organization and movement along actin bundles (*Ropars et al., 2016*). Myo10-driven filopodia are involved in processes such as neural and vascular development (*Heimsath et al., 2017*; *Pi et al., 2007*; *Zhu et al., 2007*).

Because filopodia-like structures have been observed in fusing muscles (*Gurevich et al., 2016*; *Randrianarison-Huetz et al., 2018*), we sought to determine if these structures are, indeed, Myo10-driven filopodia involved in skeletal muscle fusion. In this work, we show that Myo10, a filopodia-associated myosin, is a key component of myoblast fusion. Myo10 deficiency in myoblasts results in loss of detectable filopodia and muscle fusion in vitro and impaired muscle regeneration in vivo. Lastly, we demonstrate that the fusion proteins Myomaker and Myomixer can be detected within muscle filopodia.

## Results

### Protrusions from differentiating myoblasts are apparent during cellular fusion

Thin actin-filled cellular extensions that protrude from differentiating myoblasts have been observed during vertebrate myofusion (*Randrianarison-Huetz et al., 2018*). We sought to investigate the occurrence and behaviors of these projections in living myoblasts of both undifferentiated and differentiated states via live-cell confocal microscopy, utilizing membrane targeted fluorescent reporter constructs containing a C-terminal human H-Ras CAAX box prenylation signal (RFP-CAAX or GFP-CAAX; see Materials and methods) to enable detailed visualization of cellular protrusions. Undifferentiated myoblasts of the murine C2C12 cell line exhibit many thin cellular projections. Live imaging reveals that these projections are actively elongating primarily at the leading edge of myoblasts, while the trailing regions of the cells predominantly exhibit retraction fibers that become evident

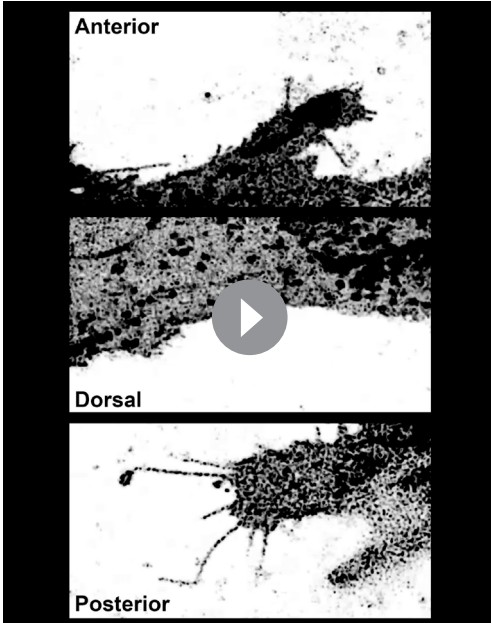

**Video 2.** Cellular projections of undifferentiated myoblasts. Representative time-lapse movies of cellular projections from the anterior, dorsal, and posterior positions of undifferentiated myoblasts expressing RFP-CAAX. Individual frames are included in Figure 2—figure supplement 1A. Images were acquired every 20 s for a duration of 2 min.

https://elifesciences.org/articles/72419/figures#video2

as the cells move (*Figure 1—figure supplement 1A-C*, *Videos 1–2*; summarized in *Figure 1A*), as commonly observed during cell migration (*Mattila and Lappalainen, 2008*). Upon induction of differentiation by switching the cells to low-serum media conditions, myoblasts undergo morphological changes characterized by cellular elongation, loss of distinct directional polarity, and increased incidence of cellular projections from the entirety of the cell body (*Video 3*). As the differentiation process proceeds to the formation of multinucleated myotubes, the cells display an array of dynamic and static projections at the lateral edges, prominent dorsal protrusions along the cell body, and arm-like lamellipodial extensions adorned with thin projections that can protrude from any part of the cell (*Figure 1B* and *Videos 4–7*; summarized in *Figure 1C*). The lengths of extending projections from myotubes are significantly longer than those from the leading edge of myoblasts (*Figure 1—figure supplement 1D*), and scanning electron micrographs confirm that these dorsal projections from myotubes are of consistent structure and size of thin, actin-based cellular projections known as filopodia (*Figure 1D*). Live imaging of differentiating myoblast cultures reveals the involvement of these structures in muscle cell fusion, as witnessed through fusion induced by projections extending from the lateral edge of myotubes (*Figure 1E–F*, *Video 8*), as well as by projection-laden lamellipodial extensions (*Figure 1G*, *Videos 9–10*). This evidence demonstrates that these cellular projections extending from differentiating myoblasts are involved in the formation of multinucleated mammalian muscle. The remainder of this report will focus on the cellular mechanisms responsible for the generation of these projections and their role in muscle fusion.

## Myo10 is required for filopodia formation and cellular fusion of myoblast in vitro

A hallmark of filopodia is the presence of Myo10, which is a molecular motor associated with the initiation and elongation of filopodia and potential cargo binding/transport within filopodia (*Berg and Cheney, 2002*; *Zhang et al., 2004*). To establish if the projections we visualize during myogenic fusion are, indeed, filopodia, we investigated the expression pattern of Myo10 within differentiating myoblast cultures. Myo10 protein content (*Figure 2A*, *Figure 2—figure supplement 1A*) and *Myo10* gene expression (*Figure 2B*) increase during the time course of myoblast differentiation. Myo10 immunofluorescence localizes specifically to differentiated myotubes (*Figure 2C*), which are confirmed to have expression of myosin heavy chain (MHC; *Figure 2D*). The Myo10-positive projections observed on these myotubes also contain actin filaments (F-actin; *Figure 2E*), a key feature of filopodia.

Evidence suggests that Myo10 can exist as an inactive and diffusible folded monomer that undergoes a conformational change during activation that allows for unfolding and anti-parallel dimer formation, resulting in engagement with the actin cytoskeleton (*Ropars et al., 2016*; *Umeki et al., 2011*). Because Myo10 appears to fill the entire cell of differentiated myotubes, we sought to determine if myocyte Myo10 represents a freely diffusible population, an actin-bound population, or combination of the two states. Fractionation of differentiating myoblast cultures into soluble and insoluble cellular fractions revealed that Myo10 partitions into both the soluble and insoluble fractions (*Figure 2—figure supplement 1B*), with a slightly larger proportion residing in the soluble fraction. Serving as fractionation controls, αTubulin partitions primarily into the soluble cellular fraction, while

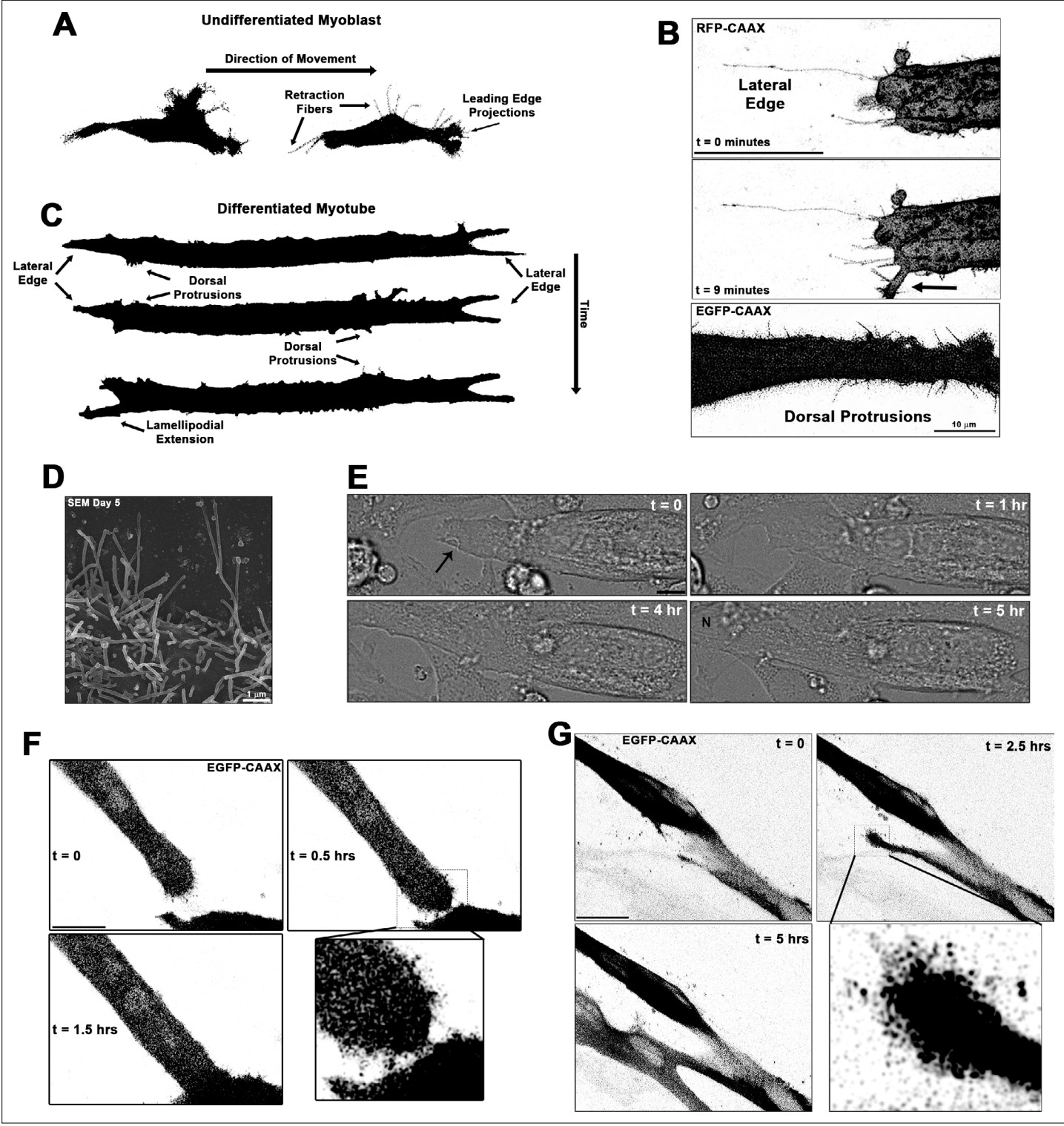

**Figure 1.** Cellular projections are prominent on differentiating muscle cells and participate in cell fusion. (**A**) A summary schematic depicting the cellular protrusions exhibited by undifferentiated myoblasts. (**B**) Live-cell confocal microscopy of differentiated myotubes (day 5) expressing a membrane-targeted fluorescent protein constructs (RFP-CAAX or GFP-CAAX) reveals myogenic projections are dynamic structures featured across the cell surface, including prominent lateral edge projections, dorsal protrusions, and those emerging from lamellipodial extensions (indicated by arrow). (**C**) Summary schematic displaying cellular projections associated with differentiated myotubes. (**D**) Cellular projections visualized on the surface of differentiating myoblasts by scanning electron microscopy (SEM). (**E**) Differential interference contrast imaging of a myotube exhibiting lateral edge protrusions (indicated by arrow) actively engaged in myoblast fusion (N indicates newly incorporated nucleus; days 4–5). Fluorescently labeled myoblasts utilizing (**F**)

*Figure 1 continued on next page*

*Figure 1 continued*

lateral edge protrusions and (**G**) a lamellipodial extension adorned with fine protrusions to promote fusion with adjacent cells (differentiation days 4–5). EGFP-CAAX in (**G**) becomes transferred to the non-expressing cell upon fusion, making the newly added cell visible via fluorescence. Unless otherwise noted, scale bars represent 25 μm.

The online version of this article includes the following figure supplement(s) for figure 1:

**Figure supplement 1.** Dynamics of myoblast cellular projections.

**Figure supplement 1—source data 1.** Source data file for *Figure 1—figure supplement 1B-C*.

**Figure supplement 1—source data 2.** Source data file for *Figure 1—figure supplement 1D*.

MHC and actin are predominantly found in the insoluble fraction (*Figure 2—figure supplement 1B*). Immunofluorescence (IF) of insoluble myotube cellular remnants following soluble fraction extraction reveals that insoluble Myo10 is associated with the actin cytoskeleton (*Figure 2—figure supplement 1C*), and can be distinctly visualized at the tips of actin bundles that appear to be within myotube filopodia (*Figure 2F*).

In agreement with Myo10 expression becoming highly activated in myoblasts during myogenic differentiation, analysis of the full-length *Myo10* promoter (*Lai et al., 2013*) revealed the presence of 14 consensus E-Box motifs (CANNTG; depicted in *Figure 2—figure supplement 1D*). These motifs are DNA elements bound by myogenic regulator factors (MRFs), such as MyoD and Myogenin, during myogenesis (*Tapscott, 2005*). Co-expression of constitutive GFP-CAAX with a mApple (RFP) construct driven by the *Myo10* promoter was used to investigate Myo10 activation in myoblasts exposed to differentiation medium for 1 day compared to those undergoing differentiation for 4 days. Activation of the *Myo10* promoter, as determined by the RFP/GFP-CAAX ratio in immunoblots, is confirmed to increase proportionally with Myo10 content as myoblast differentiation progresses (*Figure 2—figure supplement 1E-F*). Live-cell imaging early in the differentiation time course (day 1) revealed individual myoblasts with low basal expression of mApple detaching from the culture substrate, undergoing a transition into a blebbing spherical morphology with increased mApple expression, and re-attachment to the culture substrate in a morphology resembling differentiated myocytes (*Figure 2—figure supplement 1G*, *Video 11*). Consistent with MRF-mediated activation of Myo10 during myoblast differentiation, Myo10-positive mononuclear myocytes exhibit positive staining for both MyoD and Myogenin, which have both been shown to bind to the *Myo10* promoter (*Cao et al., 2006*), during the first day of differentiation (*Figure 2G*). These data indicate that *Myo10* is activated early in the differentiation period of myogenesis.

The requirement of Myo10 for the formation of muscle filopodia was investigated in *Myo10* knockdown (KD) experiments using C2C12 myoblast cell lines generated by lentiviral-mediated expression of control or *Myo10*-targeted short-hairpin RNA (shRNA) and clonal selection. *Myo10* KD results in efficient loss of *Myo10* gene expression during both growth (*Figure 2—figure supplement 1H*) and differentiation (*Figure 2—figure supplement 1I*) culture conditions, as well as reduction of Myo10 protein from both culture conditions (*Figure 3A*) and loss of Myo10 IF during differentiation (*Figure 3B*). Myogenic differentiation potential per se is not affected by loss of Myo10, as MHC protein and *Myh2* gene expression do not differ between control and *Myo10* KD lines after 5 days of differentiation (*Figure 3A*, *Figure 2—figure supplement 1I*). Loss of Myo10 does, however, have a significant effect on prevalence of cellular

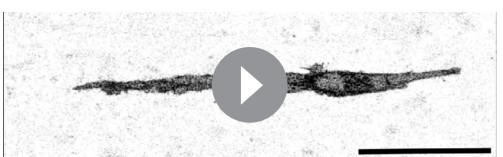

**Video 3.** Morphology changes of differentiating myoblasts. Time-lapse movie of differentiating myoblasts expressing RFP-CAAX. Images were acquired every 15 min. Scale bar represents 100 μm.
https://elifesciences.org/articles/72419/figures#video3

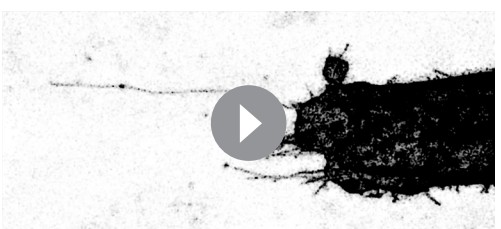

**Video 4.** Myotube lateral edge projections. Time-lapse video of the lateral edge of a differentiated myotube expressing RFP-CAAX. Individual frames are included in Figure 1B. Images were acquired every minute.
https://elifesciences.org/articles/72419/figures#video4

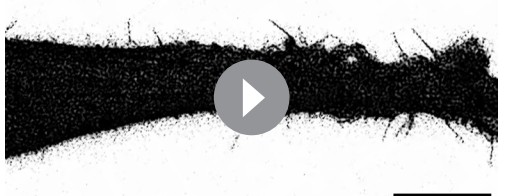

**Video 5.** Non-fluorescent detection of myotube lateral edge projections. Time-lapse of differential interference contrast imaging showing dynamic cellular projections at the lateral edge of a myotube. The arrow indicates where pronounced projections are clearly visible. Images of this myotube from later time points are also found in Figure 1E. Images were acquired every 30 s.
https://elifesciences.org/articles/72419/figures#video5

**Video 6.** Dynamic myotube dorsal protrusions. Time-lapse confocal imaging of a GFP-CAAX-expressing myotube exhibiting dynamic dorsal protrusions. Individual frames are included in Figure 1B. Images were acquired every 30 s. Scale bar represents 25 μm.
https://elifesciences.org/articles/72419/figures#video6

projections, which are therefore filopodia, as *Myo10* KD myocytes exhibit less dorsal protrusions (*Figure 3C*) and no detectable extending filopodia (*Figure 3D*; *Video 12*). The primary cellular protrusions displayed by *Myo10* KD cells appear as blebs reaching to the cell periphery in order to create connections to the surface substrate (*Video 13*), which are significantly shorter in length when compared to the thin and distinct Myo10-driven filopodia of control shRNA myocytes (*Figure 3D–E*). Loss of Myo10 does not prevent the formation of lamellipodial extensions; however, they are devoid of detectable thin projections, which are thus confirmed to be filopodia (*Video 13*).

The involvement of filopodia in myoblast fusion is also confirmed, as loss of Myo10 nearly abolishes multinucleated myotube formation following 7 days of differentiation (*Figure 3F*). This can be partially rescued by expression of exogenous Myo10, using an N-terminal mApple tagged human Myo10 construct (RFP-Myo10; *Figure 3G–H*, *Figure 2—figure supplement 1J*), as determined by the quantification of MHC-positive myocytes containing three or more nuclei following 7 days of differentiation. We chose this threshold of myonuclei content as an indication of fusion since prior studies have reported the presence of bi-nucleated myocytes following differentiation of myoblasts lacking the fusion proteins, Myomaker (*Millay et al., 2013*) or Myomixer (*Bi et al., 2017*). The rescue of *Myo10* KD myoblast fusion by exogenous Myo10 expression is not attributed to solely filopodia formation, but requires Myo10's cargo-binding functions, as a truncated RFP-Myo10 construct lacking the C-terminal PH, MyTH4, and FERM domains (RFP-Myo10ΔCBD) does not restore myoblast fusion despite promoting similar numbers of filopodia of only slightly reduced lengths (*Figure 3—figure supplement 1*). Furthermore, the requirement of myoblast Myo10 for fusogenic activity appears to be a requirement of both fusing cells, as a mixture of control and *Myo10* KD myoblasts having distinct fluorescent labels rarely results in the fusion of the two populations (*Figure 3—figure supplement 2*). Together, these data reveal that the unconventional myosin, Myo10, is important for muscle formation in vitro.

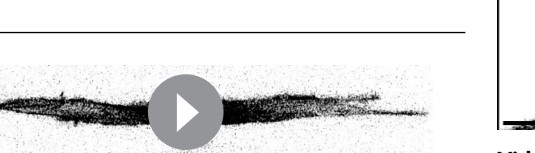

**Video 7.** Dynamics of myotube cellular projections. Time-lapse confocal imaging of diverse cellular projection of differentiating myotubes expressing GFP-CAAX. Individual frames were utilized to make Figure 1C. Images were acquired every 20 min.
https://elifesciences.org/articles/72419/figures#video7

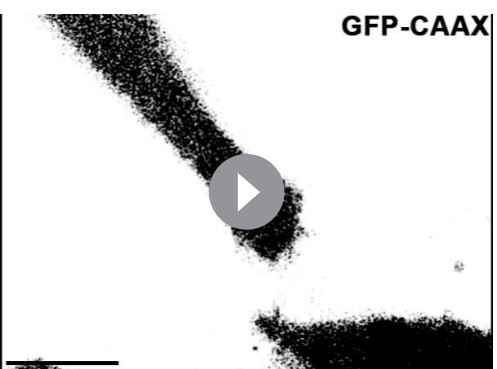

**Video 8.** Cellular fusion at the myotube lateral edge. Time-lapse confocal images of a GFP-CAAX-expressing myotube initiating fusion with fine protrusions extending from the lateral edge. Individual frames are included in Figure 1F. Images were acquired every 15 min. Scale bar represents 25 μm.
https://elifesciences.org/articles/72419/figures#video8

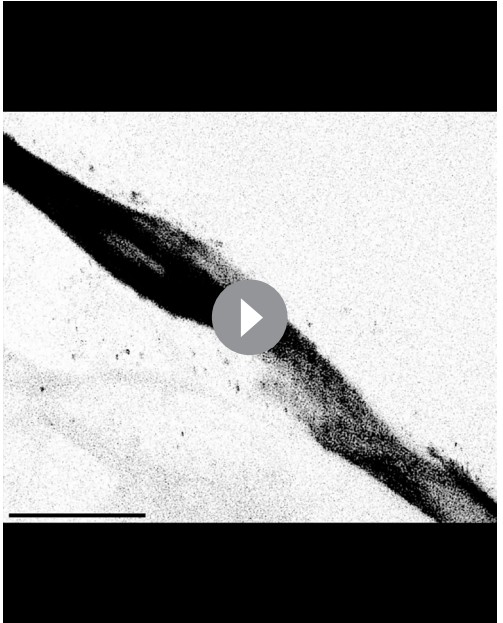

**Video 9.** Myotube fusion initiated by a lamellipodial extension. Time-lapse confocal images of a GFP-CAAX-expressing myotube initiating fusion using a lamellipodial extension adorned with fine protrusions. Individual frames are included in Figure 1G. Images were acquired every 15 min. Scale bar represents 25 µm.

https://elifesciences.org/articles/72419/figures#video9

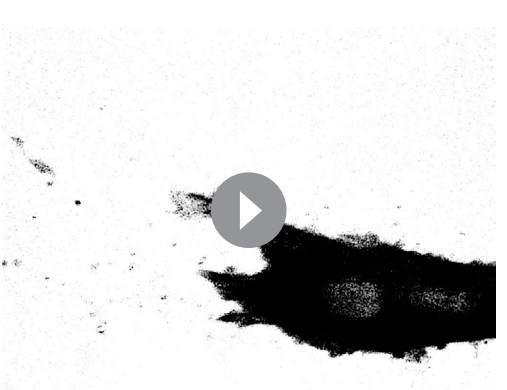

**Video 10.** Cellular fusion initiated by a lamellipodial extension. Time-lapse confocal images of a GFP-CAAX-expressing myotube initiating fusion using a lamellipodial extension. Images were acquired every 15 min.

https://elifesciences.org/articles/72419/figures#video10

## Myo10 labels regenerating muscle fibers in vivo

Given the robust impact of Myo10 loss on myoblast fusion, we next investigated Myo10-dependent skeletal muscle processes in vivo. Myo10 expression in postnatal regenerative myogenesis was examined in the muscle sections from the *mdx* mouse, a mouse model of Duchenne muscular dystrophy (DMD) that continuously displays regions of stable, damaged, and regenerating muscle fibers within the same muscle section, due to loss of dystrophin (*Hoffman et al., 1987*; *Petrof et al., 1993*). Strong Myo10 immunoreactivity within small muscle fibers of regenerating areas, which are identified by the presence of centrally located nuclei (*Coulton et al., 1988*), occurs in these muscle sections, with negligible signal detection in regions of stable muscle fibers (*Figure 4A*). This Myo10 expression pattern is also observed in human muscle, as DMD patient biopsy samples contain many small, Myo10-positive fibers localized to regenerating foci (*Figure 4B*). Thus, Myo10 is expressed in muscle during times that are expected to have high amounts of myoblast fusion, including muscle regeneration.

Since asynchronous bouts of degeneration and regeneration in parallel characterize dystrophic muscle diseases, an acute model of synchronized muscle damage and subsequent regeneration was employed in non-dystrophic mice to verify that Myo10 of muscle fibers is associated with regenerative myogenesis rather than damage or degeneration. Intramuscular injection of the myotoxin, cardiotoxin (CTX), into the tibialis anterior (TA) of *Pax7^Cre-ERT2* mice crossed with a nuclear-localized mCherry reporter (Pax7-mCherry^NLS), a model which allows for satellite cell fate-mapping, was performed. This enables distinct labeling of regenerative myogenesis by identifying mCherry-positive nuclei after tamoxifen-induced Cre-recombinase activation (*Nishijo et al., 2009*). Post-injury muscle development reveals highly elevated Myo10 levels within regenerating, mCherry-positive myofibers 4 days following CTX injection (*Figure 4C*). Analysis of muscle after 8 and 16 days of regeneration shows Myo10 content to progressively decline back to uninjured levels as the myofibers reach post-regenerative maturation (confirmed via immunoblotting; *Figure 4—figure supplement 1A*). This phenomenon is not exclusive to the CTX model, as intramuscular glycerol injection, an alternative muscle regeneration model, also results in mCherry-positive regenerating myofibers strongly labeled by Myo10 (*Figure 4—figure supplement 1B*). Therefore, regenerative myogenesis exhibits muscle-specific expression of Myo10 similar to findings in vitro. These data also demonstrate that Myo10 is an effective marker to label regenerating skeletal muscle fibers in vivo, which may be a useful tool to identify newly formed muscle fibers in lieu of developmental MHC isoforms.

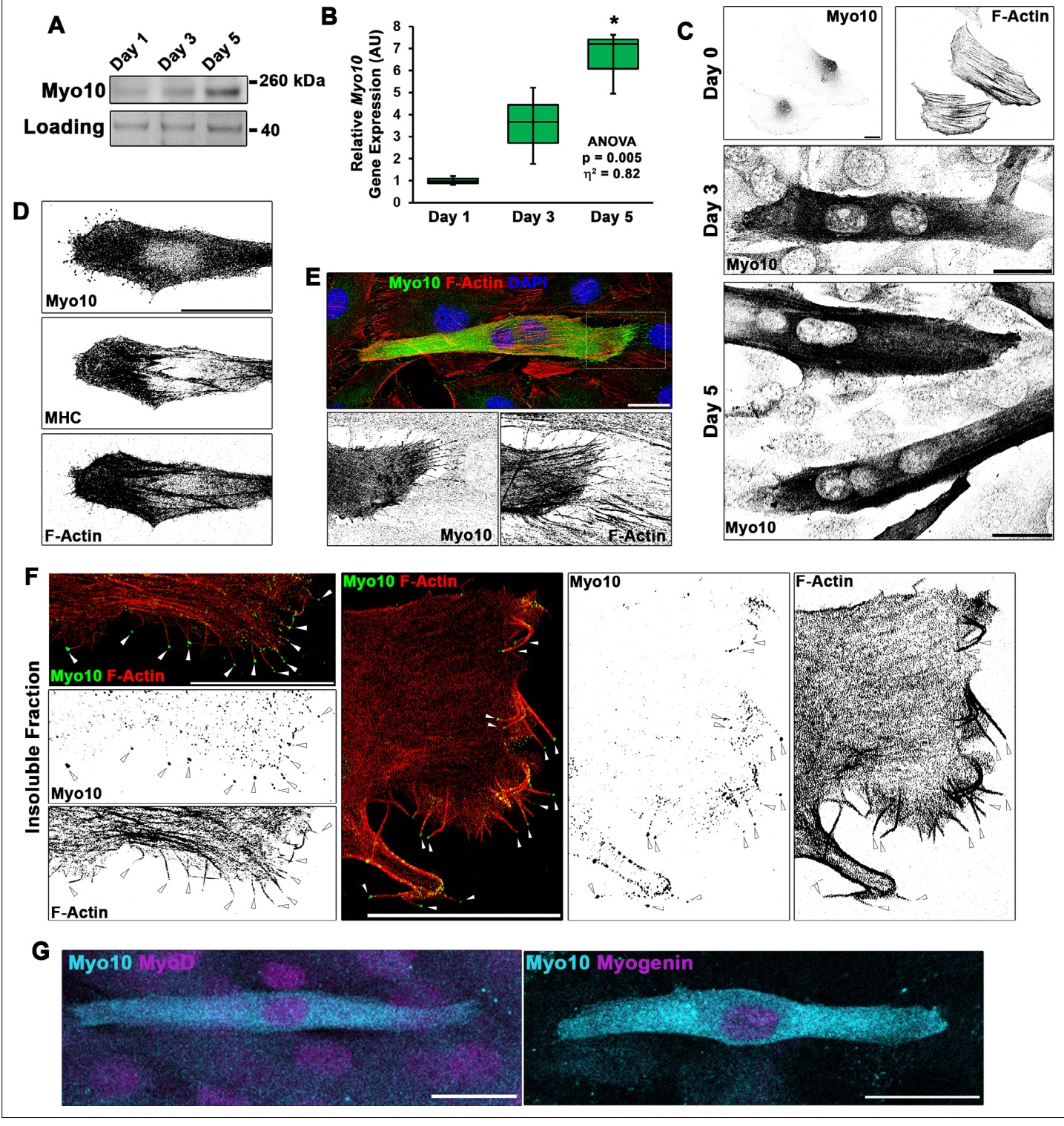

**Figure 2.** Differentiating myoblast cultures express class X myosin (Myo10). (**A**) Myo10 protein content, as shown by immunoblotting, and (**B**) *Myo10* gene expression, as measured by real-time PCR (n = 3 independent cultures for each time point; normalized to *Gapdh*), is increased during the myoblast differentiation time course. (**C**) Immunofluorescence (IF) reveals that the increase in Myo10 in differentiating myoblast cultures is localized primarily to differentiated myotubes, which (**D**) also express the muscle terminal differentiation marker, myosin heavy chain (MHC). (**E**) The Myo10-positive cellular extensions of differentiated myoblasts contain F-actin. (**F**) IF of the insoluble fraction of differentiating myoblasts reveals that insoluble Myo10 is found distinctly at the tips of F-actin bundles (indicated by arrows). (**G**) Myo10-positive myoblasts exhibit nuclear staining for both MyoD and Myogenin muscle regulatory factors, as shown by IF, following 1 day of exposure to differentiation conditions. Gene expression data are presented as

*Figure 2 continued on next page*

*Figure 2 continued*

box-and-whisker plots depicting second and third quartiles with minimum and maximum values (relative to day 1 values). Data were analyzed using one-way ANOVA followed by Tukey post hoc tests ($\alpha = 0.05$; *$p < 0.05$ vs. day 1 values; effect size is presented as eta-squared ($\eta^2$)). Unless otherwise noted, scale bars represent 25 μm.

The online version of this article includes the following figure supplement(s) for figure 2:

**Source data 1.** Source data file for *Figure 2B*.

**Figure supplement 1.** Class X myosin (Myo10) is expressed by myoblasts during muscle differentiation and is required for myoblast fusion.

**Figure supplement 1—source data 1.** Source data file for *Figure 2—figure supplement 1A*.

**Figure supplement 1—source data 2.** Source data file for *Figure 2—figure supplement 1B*.

**Figure supplement 1—source data 3.** Source data file for *Figure 2—figure supplement 1E*.

**Figure supplement 1—source data 4.** Source data file for *Figure 2—figure supplement 1F*.

**Figure supplement 1—source data 5.** Source data file for *Figure 2—figure supplement 1H*.

**Figure supplement 1—source data 6.** Source data file for *Figure 2—figure supplement 1I*.

## Loss of Myo10 in satellite cells impairs muscle regeneration

The consequence of myoblast-specific loss of Myo10 on muscle regeneration in vivo was assessed using *Pax7^{Cre-ERT2}* mice crossed to the floxed *Myo10* (*Myo10^{tm1cltm1c}*) allele (*Heimsath et al., 2017*), generating a

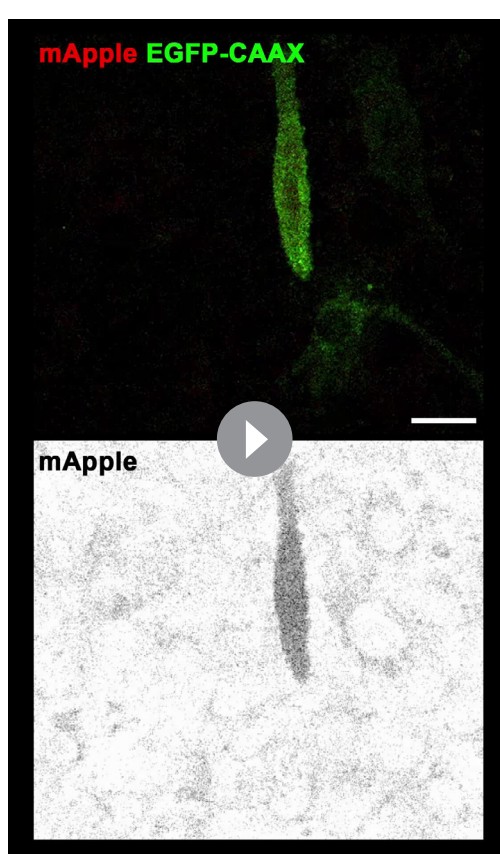

**Video 11.** Activation of the *Myo10* promoter in differentiating myoblasts. Time-lapse confocal images of a myoblast co-expressing GFP-CAAX and a reporter plasmid consisting of mApple driven by the *Myo10* promoter at day 1 of differentiation. Individual frames are included in Figure 2—figure supplement 1G. Images were acquired every 20 min. Scale bar represents 10 μm.

https://elifesciences.org/articles/72419/figures#video11

mouse line capable of inducible ablation of *Myo10* in satellite cells and, thus, their myoblast progeny and any resulting myofibers. In the absence of tamoxifen-induced *Myo10* ablation, homozygous *Myo10^{tm1c/tm1c}* mice (termed Pax7-M10cKO for Pax7-<u>M</u>yo<u>10</u> conditional <u>k</u>n<u>o</u>ck<u>o</u>ut) have indistinguishable phenotypes from their *Myo10^{tm1c/+}* or *Myo10^{+/+}* littermates (termed Pax7-WT) following CTX-induced regeneration in the TA (*Figure 4—figure supplement 1C-D*). Tamoxifen-induced Cre expression (via the protocol depicted in *Figure 4D*) results in efficient ablation of Myo10-positive muscle fibers (*Figure 4E*) and Myo10 protein content (*Figure 4—figure supplement 1E*) in Pax7-M10cKO mice 4 days following CTX injection. Remnants of extracellular matrix from pre-existing muscle fibers, known as 'ghost fibers' (*Webster et al., 2016*), predominate Pax7-M10cKO muscle sections at this time point. At 8 days following CTX injection, Pax7-WT muscles demonstrate robust regeneration, while Pax7-M10cKO muscles exhibit impaired regeneration, as evidenced by fewer and smaller myocytes present in regenerating musculature (*Figure 4F–H*). Pax7-M10cKO muscle regenerative defects are also exhibited following freeze injury, a more severe muscle injury model (*Hardy et al., 2016*), of which affected Myo10-deficient musculature is largely replaced by intramuscular fibrosis following 21 days of recovery (*Figure 4—figure supplement 2*). Thus, Myo10 is important for regenerative myogenesis.

## Muscle fusion proteins localize to filopodia

A possible role for Myo10-driven filopodia in muscle fusion is that they provide a means to

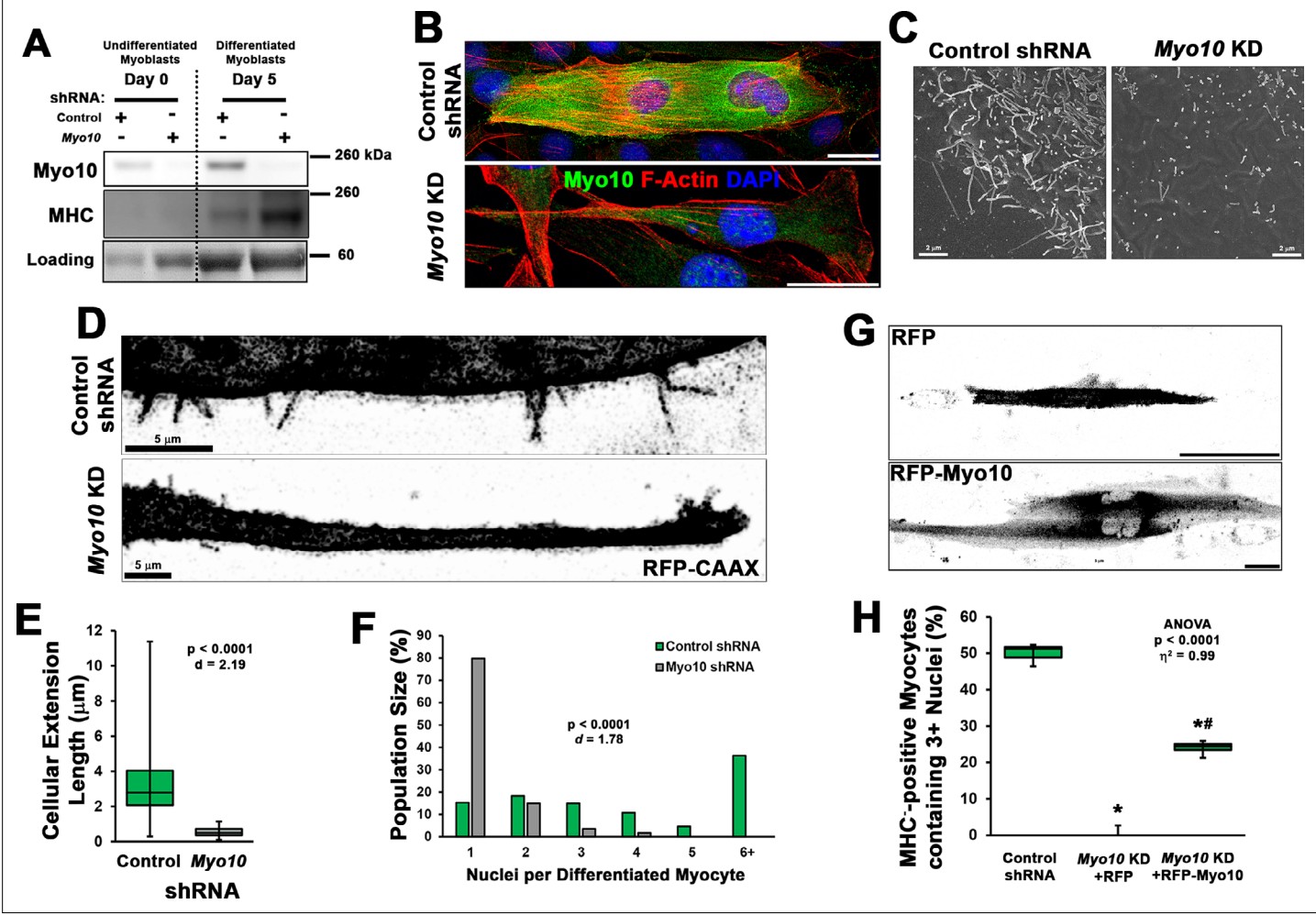

**Figure 3.** Loss of myoblast class X myosin (Myo10) prevents filopodia formation and cellular fusion. Clonal lines of C2C12 cells expressing control or *Myo10*-targeted short-hairpin RNA (shRNA) were validated for efficacy of *Myo10* knockdown (KD) and myogenic differentiation potential. (**A**) Immunoblotting for Myo10 protein and the myogenic differentiation marker, myosin heavy chain (MHC; loading control visualized by Ponceau Red staining). KD of *Myo10* myoblasts results in loss of filopodia during differentiation compared to control shRNA cells, as demonstrated by (**B**) immunofluorescence (day 3), (**C**) scanning electron microscopy (day 5), and (**D**) live-cell confocal microscopy (day 5), as well as loss of (**E**) cellular extension lengths (n = 31–152 cellular extensions). Myoblast differentiation assays (n = 3 individual experiments) reveal loss of multinucleated myotubes formation in *Myo10* KD cells after 7 days of differentiation compared to control cells, quantified as (**F**) population distribution of myotube nuclear content. (**G–H**) Loss of fusion ability by *Myo10* KD cells can be partially rescued by transfection of a full-length Myo10 construct with an N-terminal mApple fluorescent tag (RFP-Myo10; n = 3–6 individual experiments). Data analysis performed using (**E–F**) Welch's two-tailed t-test ($\alpha$ = 0.05) with effect size displayed as Cohen's d (*d*) or (**H**) one-way ANOVA followed by Tukey post hoc tests ($\alpha$ = 0.05; *p < 0.05 vs. control values; #p < 0.05 vs. RFP values; effect size is presented as eta-squared ($\eta^2$)). Unless otherwise noted, scale bars represent 25 µm.

The online version of this article includes the following figure supplement(s) for figure 3:

**Source data 1.** Source data file for *Figure 3E*.

**Source data 2.** Source data file for *Figure 3F*.

**Source data 3.** Source data file for *Figure 3H*.

**Source data 4.** Source data file for *Figure 3A*.

**Figure supplement 1.** Rescue of myoblast fusion requires the class X myosin (Myo10) cargo-binding domains.

**Figure supplement 1—source data 1.** Source data file for *Figure 3—figure supplement 1B*.

**Figure supplement 1—source data 2.** Source data file for *Figure 3—figure supplement 1C-D*.

**Figure supplement 1—source data 3.** Source data file for *Figure 3—figure supplement 1E*.

**Figure supplement 2.** Fusion of myoblasts in vitro requires bilateral class X myosin (Myo10) expression.

**Figure supplement 2—source data 1.** Source data file for *Figure 3—figure supplement 2C*.

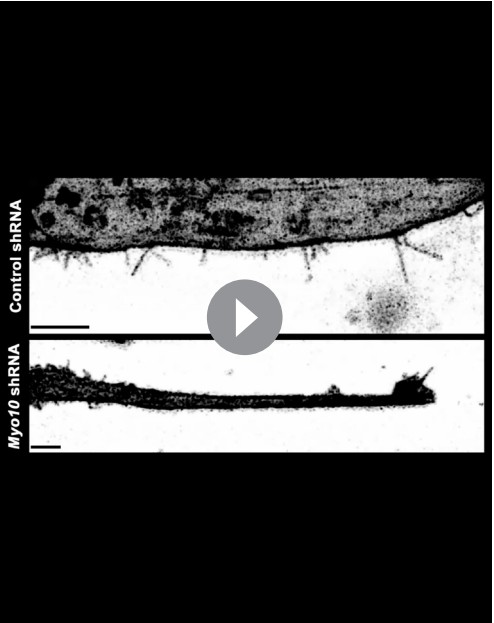

**Video 12.** Loss of class X myosin (Myo10) in myocytes prevents filopodia formation. Confocal images of cellular projections exhibited by differentiated control and *Myo10* short-hairpin RNA (shRNA) knockdown myocytes. Individual frames are included in Figure 3D. Images were acquired every 10 s. Scale bar represents 5 µm.

https://elifesciences.org/articles/72419/figures#video12

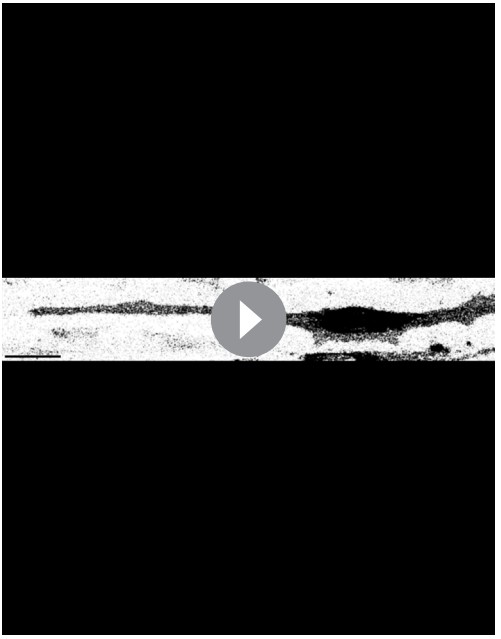

**Video 13.** Loss of class X myosin (Myo10) does not prevent lamellipodial extension formation. Differentiating Myo10 knockdown myoblasts expressing RFP-CAAX produce lamellipodial extensions during differentiation. Images were acquired every 20 min. Scale bar represents 10 µm.

https://elifesciences.org/articles/72419/figures#video13

create cellular contacts required for delivery of the fusion proteins, Myomaker and Myomixer, to apposing cellular membranes at a distance, thus increasing the probability of a fusion event occurring. To investigate this possibility, the localization of these fusion proteins on differentiating myoblasts was assessed via IF utilizing commercial antibodies for Myomixer (extracellular epitope; applied prior to cellular permeabilization) and Myomaker (intracellular epitope; applied following permeabilization). The ability of these antibodies to provide specific signals for their respective target proteins was evaluated using exogenous expression of wild-type versions of Myomaker or Myomixer in undifferentiated myoblasts (*Figure 5—figure supplement 1A-B*).

Mononuclear myocytes early in the differentiation process exhibit strong extracellular staining of Myomixer on most of the cell periphery, including cellular projections (*Figure 5—figure supplement 1C*). In these cells, Myomaker is localized primarily to vesicular structures that are particularly evident in the perinuclear cap region (*Figure 5—figure supplement 1C*). This agrees with the previously reported Golgi localization of Myomaker (*Gamage et al., 2017*). In multinucleated myotubes, Myomixer remains prominently localized at the cellular periphery, while Myomaker is additionally observed in puncta found in close proximity to the cell membrane along the cell body and cellular projections, including filopodia protruding from lamellipodial extensions (*Figure 5A*, *Figure 5—figure supplement 1D*). Myo10-positive cells co-expressing Myomaker and Myomixer are observed during the first day of differentiation (*Figure 5—figure supplement 1E*), indicating temporal regulation of these myogenic genes are synchronized. The expression of Myomixer and Myomaker are not, however, dependent on Myo10, as both proteins are found in differentiated Myo10 KD myoblasts (*Figure 5—figure supplement 1F*). In fully differentiated Myo10-positive myotubes, puncta of both Myomaker and Myomixer are observed in Myo10-filled filopodia (*Figure 5B*, *Figure 5—figure supplement 1G*). Furthermore, these proteins are found to co-localize with Myo10 puncta in the filopodia remnants of insoluble myotube fractions (*Figure 5*). The localization of both Myomaker and Myomixer to Myo10-positive filopodia is also observed when functional Flag-tagged versions of these

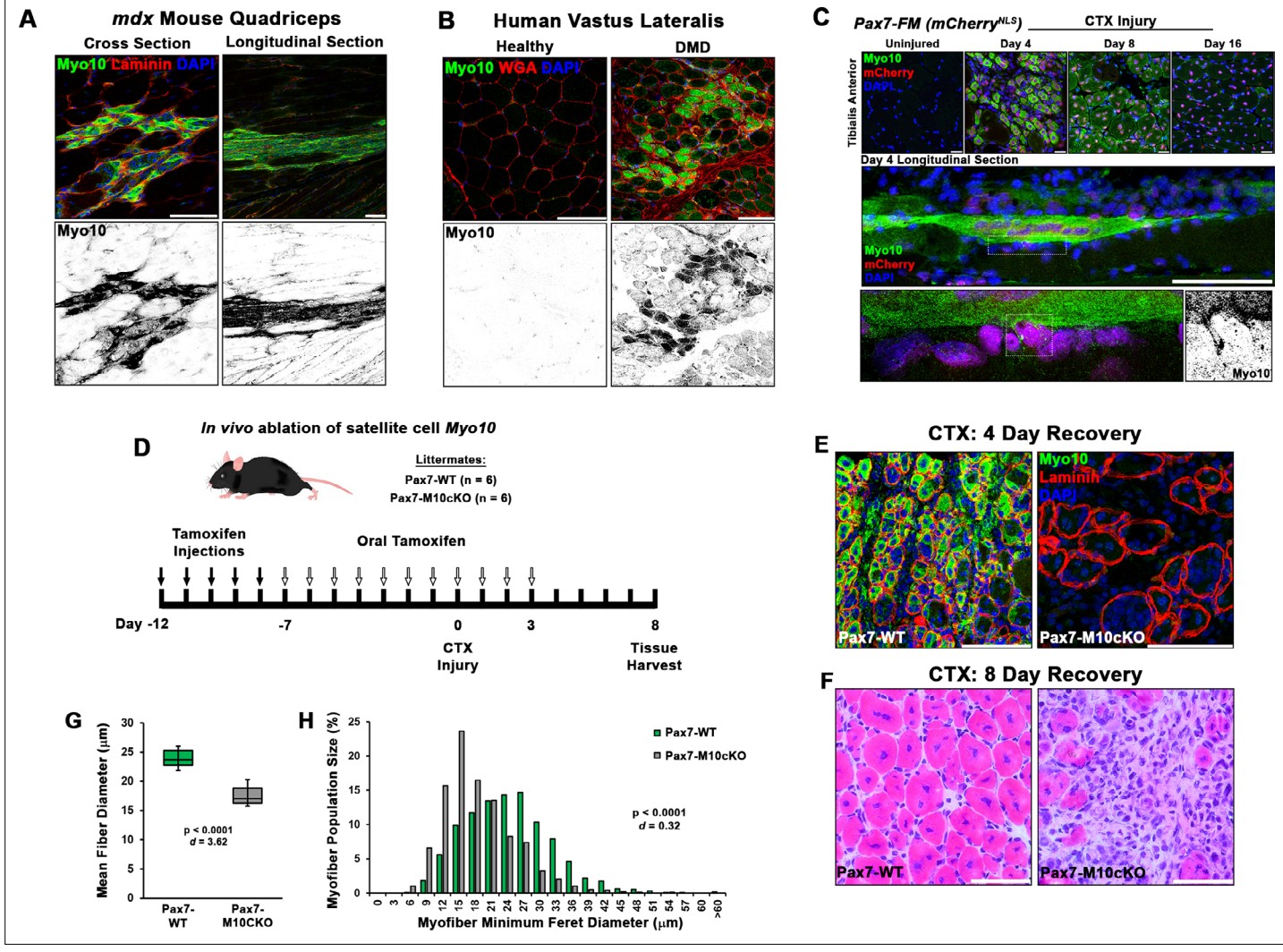

**Figure 4.** Class X myosin (Myo10) labels regenerating muscle fibers in vivo and is required for efficient muscle regeneration from injury. Myo10 immunoreactivity in regions of dystrophin-deficient skeletal muscle samples from (**A**) *mdx* mice and (**B**) Duchenne muscular dystrophy (DMD) patients that are undergoing active regeneration. (**C**) Fate-mapping (FM) of muscle satellite cells using the Pax7[Cre-ERT2] allele crossed onto mice harboring a floxed nuclear-localized (NLS) mCherry allele demonstrates that Myo10 expression is found in regenerating muscle fibers of satellite cell origin. The inset shows Myo10-filled filopodia can found extending toward mononuclear myoblasts. (**D**) The role of Myo10 in postnatal muscle regeneration was investigated using Pax7[Cre-ERT2] conditional *Myo10* knockout (KO) (Pax7-M10cKO; n = 6) mice and their non-floxed littermates (Pax7-WT; n = 6). Tamoxifen induction was achieved via five consecutive daily intraperitoneal injections of 100 mg/kg tamoxifen (denoted by solid arrows) followed by daily oral treatments with 10 mg/kg tamoxifen (empty arrows) for 7 days preceding cardiotoxin (CTX) injury of the tibialis anterior muscle (TA) and continuing until 3 days after injury. (**E**) This protocol that results in efficient elimination of Myo10+ myocytes as evidenced in 4 -day recovery muscle. Following 8 days of recovery, Pax7-M10cKO mice demonstrate impaired muscle regeneration compared to Pax7-WT muscle, as evidenced by (**F**) impaired histological recovery and (**G–H**) reduced muscle fiber size (n = 991–1307 fibers). Data are presented as (**G**) box-and-whisker plots depicting second and third quartiles with minimum and maximum values or (**H**) a histogram of entire data set populations, and are analyzed using two-tailed Welch's t-tests with effect size presented as Cohen's d (*d*). Scale bars represent 100 µm.

The online version of this article includes the following figure supplement(s) for figure 4:

**Source data 1.** Source data file for *Figure 3G–H*.

**Figure supplement 1.** Class X myosin (Myo10) is elevated during muscle regeneration.

**Figure supplement 1—source data 1.** Source data file for *Figure 4—figure supplement 1D*.

**Figure supplement 1—source data 2.** Source data file for *Figure 4—figure supplement 1A*.

**Figure supplement 1—source data 3.** Source data file for *Figure 4—figure supplement 1E*.

**Figure supplement 2.** Satellite cell ablation of class X myosin (Myo10) impairs muscle regeneration from freeze injury.

*Figure 4 continued on next page*

*Figure 4 continued*

**Figure supplement 2—source data 1.** Source data file for *Figure 4—figure supplement 2B*.

**Figure supplement 2—source data 2.** Source data file for *Figure 4—figure supplement 2C*.

proteins, namely the Myomaker-F203 (*Millay et al., 2013*) and Myomixer-Flag (*Zhang et al., 2017*) constructs, are exogenously expressed in differentiating myoblast cultures (*Figure 5—figure supplement 2*).

Sections from regenerating Pax7-WT muscle also display Myo10-labeled filopodia decorated with Myomixer puncta (*Figure 5E*), suggesting this relationship also exists in vivo. Such cellular projections are not found on Myomaker- and Myomixer-positive cells of regenerating Pax7-M10cKO muscle (*Figure 5F*), which exhibit thin morphologies similar to *Myo10* KD myoblasts rather than robust myotube formation found in wild-type muscle sections (*Figure 4C*). These findings are consistent with the hypothesis that muscle filopodia provide a means to enable cellular connections that facilitate the function of these fusion proteins upon delivery to a cellular target.

## Discussion

Cellular projections provide cells the ability to explore surrounding space and present molecules for intercellular communication and interaction during tissue development and regeneration. The current work demonstrates that the finger-like projections observed on differentiating mammalian myoblasts are Myo10-driven filopodia that participate in myoblast fusion. Data also demonstrate that myoblast expression of Myo10 is required for formation of multinucleated myotubes in vitro and efficient regeneration of skeletal muscle from injury in vivo. Furthermore, evidence is provided demonstrating that the fusogenic proteins Myomaker and Myomixer are observed on Myo10-powered filopodia.

Myosin motors compose a diverse protein superfamily with many classes that have all evolved to perform specialized cellular processes. With roles encompassing myocyte contractility, cellular movement, vesicle transport, endocytosis, organelle positioning, and formation and maintenance of filopodia, stereocilia, and microvilli (*Sweeney and Holzbaur, 2018*), the myosins expressed in humans are essential for many physiological activities. This is highlighted by a large number of diseases resulting from myosin mutations, including cardiomyopathy (*Geisterfer-Lowrance et al., 1990*; *Mohiddin et al., 2004*), skeletal myopathy (*Armel and Leinwand, 2009*), deafness (*Friedman et al., 1999*; *Mohiddin et al., 2004*), aneurysms (*Pannu et al., 2007*), and enteropathy (*Golachowska et al., 2012*).

While the involvement of the actin cytoskeleton is well documented in both arthropod and mammalian myoblast fusion (*Chen, 2011*; *Millay et al., 2013*; *Peckham, 2008*; *Randrianarison-Huetz et al., 2018*; *Segal et al., 2016*), no motor proteins have been previously described to have a direct role in the fusion process itself. Only the conventional myosin, non-muscle myosin II, has been implicated in myoblast fusion, serving as a sub-sarcolemmal mechanosensor (*Kim et al., 2015*). The current report also provides the first described role of Myo10 in mammalian muscle, which has only been previously detected in muscle via dystrophic muscle gene expression arrays (*Marotta et al., 2009*) and *Myo10* promoter binding by the muscle regulatory factors MyoD and Myogenin (*Cao et al., 2006*).

Interestingly, a myosin motor involved in myoblast fusion has not been identified in *Drosophila*, which lack Myo10. It is likely that another member of MyTH4-FERM containing myosins, such as the class XV myosin (Myo15) homolog *Sisyphus* (*Liu et al., 2008*), is adapted for the role of filopodia formation in insect muscle. A recent study has demonstrated that loss of Myo15 in *Drosophila* substantially reduces larva viability and causes abnormal neuromuscular junction formation, whereas muscle-specific Myo15 overexpression causes the development of peculiar F-actin structures in myocytes (*Rich et al., 2021*). While this report did not address whether these actin-based structures are associated with cellular protrusions, the images provided do resemble thin actin bundles reminiscent of myotube dorsal filopodia described in the current study. While the definitive identification of a myosin motor that drives insect muscle filopodia awaits further investigation, the premise of filopodia-facilitated myoblast fusion appears to be a convergent evolutionary feature of multinucleated muscle formation (*Segal et al., 2016*). Thus, the physical presentation

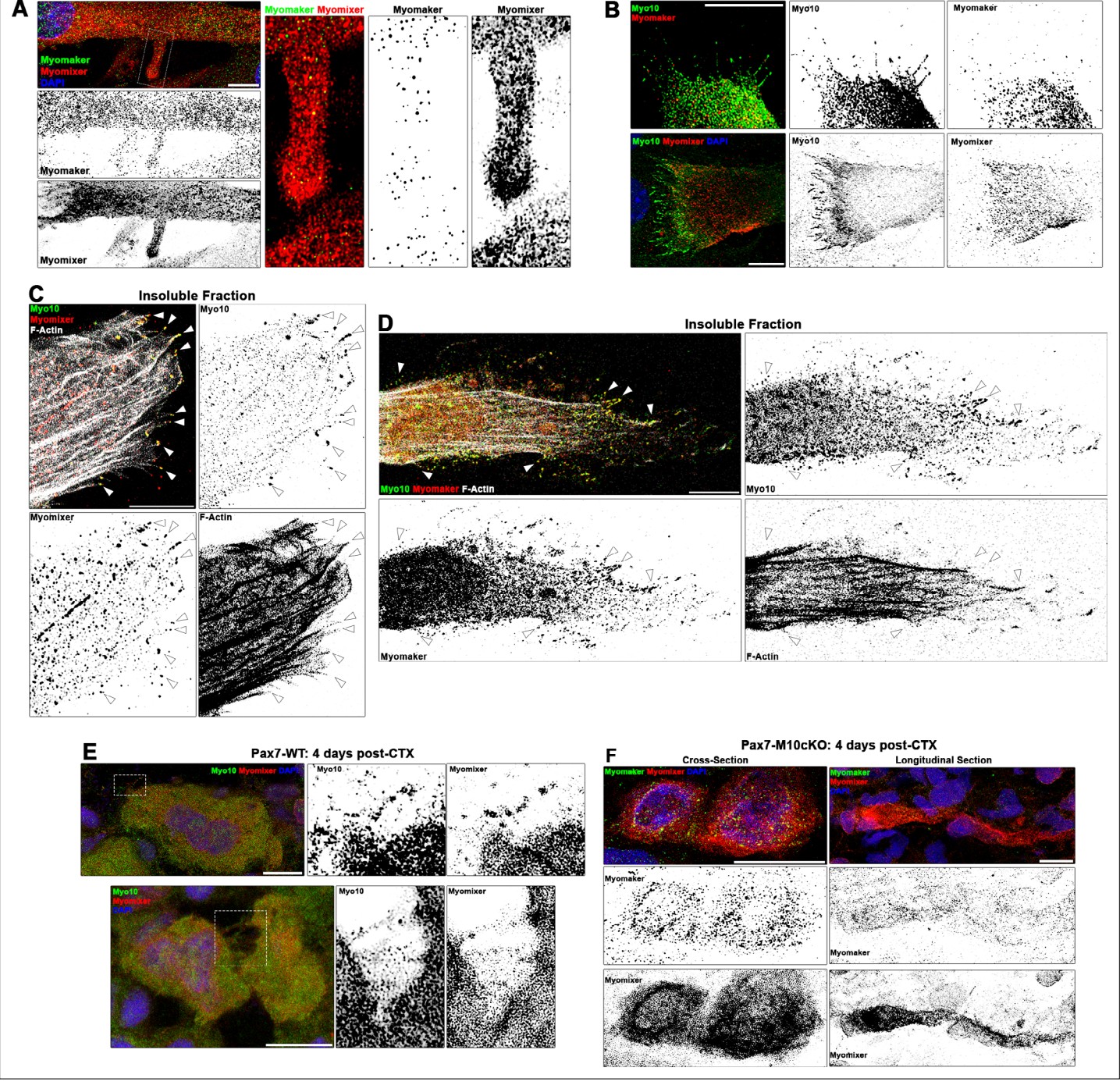

**Figure 5.** Myogenic fusion proteins are detected along filopodia in differentiating muscle cells. (**A**) Immunofluorescent detection of Myomaker and Myomixer in differentiating myoblast cultures. Inset shows Myomixer and Myomaker puncta on the surface of a lamellipodial extension with filopodia. (**B**) Myomaker and Myomixer puncta are found localized to class X myosin (Myo10)-filled filopodia of differentiating myoblasts (days 4–5). Co-localization of (**C**) Myomixer and (**D**) Myomaker with Myo10 in filopodia remnants of the differentiating myoblast insoluble fraction (indicated with arrows). (**E**) Tibialis anterior (TA) muscle cross-sections of Pax7-WT mice at 4 days following cardiotoxin (CTX)-induced injury. (**F**) Cross-section (left) and longitudinal section (right) of Pax7-M10cKO TA muscle at 4 days following CTX-induced injury. Scale bars represent 10 μm.

The online version of this article includes the following figure supplement(s) for figure 5:

**Figure supplement 1.** Localization patterns of Myomaker and Myomixer in differentiating myoblasts.

**Figure supplement 2.** Localization patterns of Flag-tagged Myomaker and Myomixer constructs in differentiating myoblasts.

of molecules at a distance from the cell body via cytoskeletal extensions is an important aspect of skeletal muscle development as it is in the development of other tissue types (*Pi et al., 2007*; *Zhu et al., 2007*).

A paramount finding of this report is the defective muscle regeneration caused by loss of Myo10 in satellite cells. These data are potentially of clinical significance, as a patient having two alleles for a truncating *MYO10* mutation has been recently identified (*Patel et al., 2018*). While this individual was found on basis of having microphthalmia, a developmental eye defect, it is possible that an underlying myopathy may exist or develop as a result of impairments in muscle recovery from injury. In fact, microphthalmia is also associated with several syndromes that also present with myopathy or neuromuscular disorders, including Walker-Walburg syndrome (*Vajsar and Schachter, 2006*) and Charcot-Marie-Tooth disease (*Fernandez-Torre et al., 2001*). Furthermore, several *MYO10* single-nucleotide polymorphisms have been identified (*Burghardt et al., 2010*), including missense mutations in the N-terminal motor domain. Mutations such as these have the potential to create dominant-negative Myo10 molecules if motor function is impaired or destroyed, and individuals harboring such polymorphisms may exhibit muscle regenerative defects, as demonstrated by loss of *Myo10* in mice. The identification of a Myo10-deficient patient also confirms findings in mice that loss of Myo10 is not absolutely lethal, although less than half of Myo10-null mice survive birth, and those that do survive exhibit developmental deficits (*Heimsath et al., 2017*). These evidences of decreased Myo10-null embryonic survivability further indicate that Myo10-driven filopodia serve to increase the probability of proper cellular connectivity during developmental processes. The fact that the surviving Myo10-null animals have fused muscle fibers likely indicates that myoblasts are in close opposition during development, lessening the dependence on distal interactions mediated by filopodia. In contrast, satellite cell-derived myoblasts of injured adult muscles must traverse much larger distances in order to locate and fuse with each other, creating a greater dependence on filopodia for postnatal muscle repair than for embryonic muscle development.

Following confirmation that Myo10-driven filopodia are involved in the fusion events required to form multinucleated muscle, we investigated if there is a relationship between these fine cellular projections and the recently discovered muscle fusion proteins, Myomaker and Myomixer. Our experiments reveal that Myomaker and Myomixer are both highly expressed in Myo10-positive myocytes and present in muscle filopodia. It is currently hypothesized that Myomaker's role in cellular fusion is the promotion of outer membrane leaflet mixing (i.e. hemifusion) between cells, while Myomixer acts as an inducer of intercellular pore formation by promoting positive spontaneous membrane curvature within hemifusion structures (*Golani et al., 2021*; *Leikina et al., 2018*). These actions are consistent with the concept of their delivery by Myo10-driven filopodia in order to create an intermediate structure that is essentially a tunneling nanotube, whether configured as individual or bundled lumens (*Sartori-Rupp et al., 2019*), that precedes development of a full syncytium. Such a structural organization is suggested by the images provided in *Figure 1F*, and Myo10-dependent formation of tunneling nanotubes has been suggested as a requirement for osteoclast differentiation into multinucleated cells (*Tasca et al., 2017*). While it remains to be confirmed if Myo10 plays an active role in the localization and/or activity of these fusogenic proteins or whether Myo10-driven filopodia increase the probability proper cellular connectivity via increased surface area, the failure of a C-terminal Myo10 truncation (RFP-Myo10ΔCBD) to restore myoblast fusogenic activity, despite promoting filopodia formation, suggests Myo10 does have an active cargo-binding role in promoting fusion of mammalian myoblasts. Furthermore, the apparent requirement of Myo10 by both fusing cells, a feature also exhibited by Myomaker (*Quinn et al., 2017*), suggests that Myomaker fusogenic activity may depend on filopodia driven by full-length (cargo-binding competent) Myo10.

The findings detailed in this report describe a role of filopodia driven by the unconventional myosin, Myo10, in the formation of multinucleated skeletal muscle. These data further emphasize the importance of these fine cellular projections in the intricate biological processes required for proper development of higher-order organisms, and that perturbations to their formation and function have the ability to cause muscle pathology and potentially modify the course of muscle disease.

# Materials and methods

**Key resources table**

| Reagent type (species) or resource | Designation | Source or reference | Identifiers | Additional information |
|---|---|---|---|---|
| Strain, strain background (*Mus musculus*) | H2B-mCherry^fl/fl | Jackson Laboratories | JAX:023139 RRID:IMSR_JAX:023139 | |
| Strain, strain background (*Mus musculus*) | Pax7^Cre=ERT2 | PMID:26792330 | MGI:4436914 | |
| Strain, strain background (*Mus musculus*) | Myo10^tm1c | PMID:29229982 | MGI:6115837 | |
| Cell line (*Mus musculus*) | C2C12 | ATCC | ATCC No. CRL-1772 RRID:CVCL_0188 | |
| Transfected construct (*Homo sapiens*) | tagRFPt-HRAS-CAAX (RFP-CAAX) | This paper | Evrogen # FP141 NCBI NP_005334 | Membrane-targeted RFP |
| Transfected construct (*Homo sapiens*) | EGFP-HRAS-CAAX (GFP-CAAX) | This paper | Clontech# 632,470 NCBI NP_005334 | Membrane-targeted GFP |
| Transfected construct (*Homo sapiens*) | RFP-Myo10 | This paper | Addgene No. 54,631 NCBI No. NP_036466 | Full-length Myo10 |
| Transfected construct (*Homo sapiens*) | RFP-Myo10ΔCBD | This paper | Addgene No. 54,631 NCBI No. NP_036466 | Myo10 (aa 1–938) |
| Transfected construct (*Mus musculus*) | pMyo10-mApple | This paper | Addgene No. 54,631 | Reporter for Myo10 promoter activation |
| Transfected construct (*Mus musculus*) | Myomaker | PMID:26858401 | NCBI No. NP_079652 | |
| Transfected construct (*Mus musculus*) | Myomixer | PMID:29581287 | NCBI No. NP_001170939 | |
| Transfected construct (*Mus musculus*) | Myomaker-F203 | PMID:26858401 | NCBI No. NP_079652 | Flag-tagged Myomaker |
| Transfected construct (*Mus musculus*) | Myomixer-Flag | PMID:28569745 | NCBI No. NP_001170939 | Flag-tagged Myomixer |
| Genetic reagent (*Mus musculus*) | Control shRNA lentiviral particles | Sigma-Aldrich | No. SHCLNV shRNA ID- SHC002 | |
| Genetic reagent (*Mus musculus*) | Myo10 shRNA lentiviral particles | Sigma-Aldrich | No. SHCLNV shRNA IDs- TRCN0000110606 TRCN0000375033 | |
| Antibody | Anti-Myo10 (Rabbit polyclonal) | Sigma-Aldrich | HPA024223 | IF: (0.3 µg/mL) IB: (0.24 µg/mL) |
| Antibody | Anti-Myo10 (Mouse monoclonal) | Santa Cruz | sc166720 | IF: (2 µg/mL) |
| Antibody | Anti-Laminin (Rat monoclonal) | Acris Antibodies | BM6064P | IF: (1.25 µg/mL) |
| Antibody | Anti-Myosin Heavy Chain (Mouse monoclonal) | R&D Systems | MAB4470 | IF: (0.25 µg/mL) IB: (0.125 µg/mL) |
| Antibody | Anti-mCherry (Chicken polyclonal) | Novus | NBP2-25158 | IF: (1:2000) |
| Antibody | Anti-MyoD (Mouse monoclonal) | Thermofisher | MA1-41017 | IF: (5 µg/mL) |
| Antibody | Anti-Myogenin (Mouse monoclonal) | Novus | NB100-56510 | IF: (2 µg/mL) |

*Continued on next page*

*Continued*

| Reagent type (species) or resource | Designation | Source or reference | Identifiers | Additional information |
|---|---|---|---|---|
| Antibody | Anti-TMEM8C/ Myomaker (Rabbit polyclonal) | Thermofisher | PA5-63180 | IF: (1 µg/mL) |
| Antibody | Anti-Myomixer (Sheep polyclonal) | R&D Systems | AF4580 | IF: (0.67 µg/mL) |
| Antibody | Anti-Flag (Mouse monoclonal) | Sigma-Aldrich | F3165 | IF: (1.6 µg/mL) |
| Antibody | Anti-RFP (Rabbit polyclonal) | Abcam | ab62341 | IB: (0.5 µg/mL) |
| Antibody | Anti-GFP (Chicken polyclonal) | Abcam | ab13970 | IB: (2.5 µg/mL) |
| Sequence-based reagent (*Mus musculus*) | Myo10 Forward Primer | This paper | NCBI No. NM_019472 | TTC CAC CGC ACA TCT TCG CCA TTG |
| Sequence-based reagent (*Mus musculus*) | Myo10 Reverse Primer | This paper | NCBI No. NM_019472 | CCC CGG GAT TCT GCC TCA CTA CTC |
| Sequence-based reagent (*Mus musculus*) | Myh2 Forward Primer | This paper | NCBI No. NM_001039545 | AGA ACA TGG AGC AGA CCG TG |
| Sequence-based reagent (*Mus musculus*) | Myh2 Reverse Primer | This paper | NCBI No. NM_001039545 | TCA TTC CAC AGC ATC GGG AC |
| Sequence-based reagent (*Mus musculus*) | Gapdh Forward Primer | This paper | NCBI No. BC023196 | AGC AGG CAT CTG AGG GCC CA |
| Sequence-based reagent (*Mus musculus*) | Gapdh Reverse Primer | This paper | NCBI No. BC023196 | TGT TGG GGG CCG AGT TGG GA |

## Animals

All animal procedures were approved and conducted in accordance with the University of Florida IACUC. C57BL/10 (RRID:IMSR_JAX:000476), *mdx* (RRID:IMSR_JAX:001801), and H2B-mCherry$^{fl/fl}$ (RRID:IMSR_JAX:023139) mice used for this study were from colonies originally derived from Jackson Laboratories. *Pax7$^{Cre-ERT2}$* mice were a generous gift from Dr Charles Keller (*Nishijo et al., 2009*). The *Myo10$^{tm1c}$* floxed allele was generated as previously described (*Heimsath et al., 2017*). Tamoxifen-induced Cre expression for fate-mapping experiments was achieved by intraperitoneal injections of 20 mg/mL tamoxifen (Sigma-Aldrich No. T5648) dissolved in sterilized sunflower seed oil (Sigma-Aldrich No. S5007) at a dose of 100 mg/kg for 5 consecutive days, which results in ~85–90% labeling efficiency in Pax7 cells. To achieve near 100 % induction for conditional ablation studies, mice were subjected to the 5 -day injection protocol described above followed by daily oral administration of 10 mg/kg tamoxifen (sunflower seed oil vehicle) starting at 7 days preceding injury to 3 days following injury (depicted in *Figure 4D*). Oil-only injections and oral treatments served as sham induction controls. Injections and treatments were performed within 2 hr of the start of the mouse dark cycle to facilitate drug distribution. Only male mice were used for these experiments. The genotypes of all mice used for this study were verified by PCR-based genotyping. Mice were randomly assigned into experimental groups prior to experiments.

Injury of the TA muscle was performed by injecting 50 µL of sterile solutions of either 12 µM CTX (Calbiochem No. 217503; dissolved in sterile PBS) or 50 % glycerol longitudinally through the length of the muscle. Freeze injury was performed by applying a liquid $N_2$-cooled metal rod to the mid-belly of a surgically exposed TA from a randomly selected hind-limb for 10 s. Following the allocated recovery time from injury, mice were euthanized via $CO_2$. TA muscles were dissected free, either snap-frozen in liquid $N_2$ or embedded in OCT compound and frozen in melting isopentane, and stored at –80 °C until analysis.

## Cell culture

C2C12 murine myoblasts (ATCC No. CRL-1772; RRID:CVCL_0188; verified to be free of mycoplasma contamination) were purchased from ATCC and used between passages 5 and 13. Cells were cultured

at 37 °C in 5 % $CO_2$ in growth media consisting of high-glucose DMEM (Gibco No. 10566), 10 % fetal bovine serum (Sigma-Aldrich No. F8067), and 1 % penicillin/streptamycin (P/S; Gibco No. 15140). C2C12 differentiation media consisted of low-glucose DMEM (Gibco No. 11885), 2 % horse serum (Hyclone No. SH30074), and 1% P/S and was changed every 2 days during differentiation experiments. Ectopic expression experiments were performed using X-tremeGENE 9 DNA transfection reagent (Sigma-Aldrich No. 6365779001) or electroporation using the 4D-Nucleofector system (Lonza). For live-cell imaging, cells were plated on collagen- or gelatin-coated glass-bottom dishes (Willco Wells No. GWST-3522) prior to transfection, differentiated in phenol red-free media (using glutamate-supplemented Gibco No. 11,054 DMEM in place of No. 11885), and mounted in a stage-top incubator (Tokai HIT No. INUB-GSI2-F1) with 5 % $CO_2$ for image acquisition.

Control or *Myo10* shRNA KD C2C12 lines were made using MISSION shRNA lentiviral particles (Sigma-Aldrich No. SHCLNV; Control shRNA ID- SHC002; *Myo10* shRNA IDs- TRCN0000110606 and TRCN0000375033; 5 MOI) following the manufacturer's directions. Puromycin-resistant clones were selected and verified for *Myo10* KD efficiency and myogenic differentiation capacity. For all assays comparing control and *Myo10* KD lines, equal cell numbers were plated and switched to differentiation medium 12–16 hr after plating.

## Plasmids

The tagRFPt-HRAS-CAAX (RFP-CAAX) plasmid was generated by adding the C-terminal 20 amino acids of human H-Ras (NCBI Accession No. NP_005334) to a tagRFP-C vector (Evrogen) modified with an S158T point mutation to enhance photostability (*Shaner et al., 2008*). EGFP-HRAS-CAAX (GFP-CAAX) was constructed similarly by placing the HRAS-CAAX box motif on the C-terminus of EGFP in the pEGFP-C1 vector (Clontech). The RFP-Myo10 construct was prepared by cloning mApple (from Addgene No. 54631) to the N-terminus of human Myo10 (NCBI Accession No. NP_036466) using a G-G-R linker, similar to as previously described (*Ropars et al., 2016*), in pCDNA3.1(+) vector (Thermofisher No. V79020). The RFP-Myo10ΔCBD construct was prepared by fusing mApple to the N-terminus of a human Myo10 construct lacking the PEST, PH, MyTH4, and FERM domains (aa 1–938), as previously described (*Ropars et al., 2016*), in pCDNA3.1(+) vector. The *Myo10* reporter plasmid was constructed by replacing CMV promoter of pCDNA3.1(+) with the –835/+314 region of the *Myo10* promoter region *Lai et al., 2013* followed by an mApple open-reading frame. Murine Myomaker (NCBI Accession No. NP_079652), Myomixer (NCBI Accession No. NP_001170939), Myomaker-F203 (*Millay et al., 2013*), and Myomixer-Flag (*Zhang et al., 2017*) open-reading frames were cloned into pCDNA3.1(+) vector. All constructs were verified by sequencing and restriction analysis, and all plasmids were prepared in endotoxin-free conditions.

## IF and fluorescent labeling

Tissue IF was performed as previously described (*Hammers et al., 2016*). Briefly, OCT-embedded frozen muscle was sectioned into either cross-sections or longitudinal sections of 10 µm thickness, fixed in ice-cold acetone, blocked in 5 % BSA-PBS +0.1 % Triton X-100, and incubated in primary antibody overnight at 4 °C. Secondary antibodies were applied for 1 hr at room temperature the following day. Lipofuscin-induced autofluorescence was eliminated using 0.1 % Sudan Black B dissolved in 70 % ethanol, ensued by a wash in 0.1 % Triton X-100 in PBS. Sections were mounted in Vectashield (+DAPI; Vector Labs No. H1200), cover-slipped, and sealed. Control and DMD patient samples were acquired from the National Disease Research Interchange (NDRI; Philadelphia, PA).

Cells cultured on gelatin or collagen-coated coverslips were rinsed twice with PBS, fixed in 4 % PFA-PBS for 30 min at room temperature, permeablized with 0.1 % Triton X-100 in 4 % PFA-PBS, blocked with 0.5 % BSA-PBS, and incubated in primary antibody overnight at 4 °C. External epitopes of Myomixer were specifically stained by incubation of appropriate primary antibodies for 1 hr prior to permeabilization step. Secondary antibodies were applied for 1 hr at room temperature the following day, followed by 20 min incubation with Alexa 647-conjugated phalloidin (1:300 in PBS; Life Technologies No. A22287), when appropriate. Cells were counterstained with DAPI and mounted onto pre-cleaned glass slides with Prolong Gold mounting media (Life Technologies No. P36934). Insoluble myotube fractions were prepared for IF by removing the soluble cellular fraction via incubation of cells with ice-cold PBS containing 1 % Triton X-100, 5 mM EDTA, and protease and phosphatase inhibitor

cocktails for 1 hr. The remaining insoluble fraction of the cells was fixed in 4 % PFA following careful removal of the soluble fraction and two washes with ice-cold PBS.

Primary antibodies used for IF include anti-Myo10 (0.3 µg/mL; Sigma No. HPA024223; RRID:AB_1854248), anti-Myo10 (2 µg/mL; Santa Cruz Biotechnology No. sc166720; RRID:AB_2148054), anti-Laminin (1.25 µg/mL; Acris Antibodies No. BM6064P), anti-MHC (0.25 µg/mL; R&D Systems No. MAB4470; RRID:AB_1293549), anti-mCherry (1:2000; Novus No. NBP2-25158; RRID:AB_2636881), anti-MyoD (5 µg/mL; Thermofisher No. MA1-41017; RRID:AB_2282434), anti-Myogenin (2 µg/mL; Novus No. NB100-56510; RRID:AB_838604), anti-TMEM8C/Myomaker (1 µg/mL; Thermofisher No. PA5-63180; RRID:AB_2648742), anti-Myomixer/Myomerger/ESGP (0.67 µg/mL; R&D No. AF4580; RRID:AB_952042), and anti-Flag (1.6 µg/mL; Sigma-Aldrich No. F3165; RRID:AB_259529). Secondary antibodies (all 1:500 dilution) used include Alexa 488 donkey anti-rabbit IgG (Life Technologies No. A21206), Alexa 647 donkey anti-rabbit IgG (Life Technologies No. A31573), Alexa 568 goat anti-mouse IgG (Life Technologies No. A11031), Alexa 568 donkey anti-sheep IgG (Life Technologies No. A21099), and TRITC donkey anti-chicken IgY (Jackson No. 703-025-155). Appropriate primary antibody isotype controls were used in combination of secondary antibodies to ensure specificity of signal. All images were acquired with a Leica SP8 confocal microscope and processed with Leica LAS X software. Image acquisition was performed in sequential scan mode to ensure fidelity of fluorescent signal observed. Cellular projection lengths were analyzed from time-lapse image Z-stacks, where each projection was measured at its longest observed length using FIJI image analysis software (NIH). Image-based quantifications were performed by investigators blind to experimental groups.

## Immunoblotting

Preparation of muscle protein homogenates was performed as previously described (*Hammers et al., 2017*). Cell lysates for direct immunoblotting experiments were prepared by lysis of cell cultures with SDS-supplemented T-Per lysis reagent (Thermofisher No. 78510) containing protease and phosphatase inhibitor cocktails. For cell fractionation experiments, the soluble cellular fraction was obtained by incubation of cells with ice-cold PBS containing 1 % Triton X-100, 5 mM EDTA, and protease and phosphatase inhibitor cocktails. Following removal of this soluble fraction, the insoluble fraction was solubilized using an equal volume of SDS-supplemented T-Per lysis reagent containing protease and phosphatase inhibitor cocktails.

All samples were prepared for SDS-PAGE by boiling in Laemeli's sample buffer containing 50 mM DTT, run on 4–12% Tris-glycine SDS gels, and transferred to nitrocellulose membranes, as previously described (*Hammers et al., 2017*). Following blocking in 5 % BSA-TBST, membranes were incubated with anti-Myo10 (0.24 µg/mL; Sigma No. HPA024223; RRID:AB_1854248), anti-Myo10 (2 µg/mL; Santa Cruz Biotechnology No. sc166720; RRID:AB_2148054), anti-MHC (0.125 µg/mL; R&D Systems No. MAB4470; RRID:AB_1293549), anti-Actin (1 µg/mL; Sigma-Aldrich No. A3853; RRID:AB_262137), anti-αTubulin (1:2000; Cell Signaling No. 2144; RRID:AB_2210548), anti-RFP (0.5 µg/mL; Abcam No. ab62341; RRID:AB_945213), or anti-GFP (2.5 µg/mL; Abcam No. ab13970; RRID:AB_300798) primary antibody overnight at 4 °C. Membranes were incubated with HRP-conjugated anti-rabbit IgG (1:1000; Cell Signaling No. 7074), anti-mouse IgG (1:1000; Cell Signaling No. 7076), or anti-chicken IgY (1:1000; Jackson Labs No. 303-035-003) secondary antibody for 1 hr, and developed with ECL reagent (Thermofisher No. 34577). Images were captured using the C-Digit Imaging System (Licor). Ponceau Red staining was used to verify equal loading of comparative samples.

## Real-time PCR

Real-time PCR was performed as previously described (*Hammers et al., 2017*) using the following mouse-specific primers: *Myo10* (forward) 5'-TTC CAC CGC ACA TCT TCG CCA TTG-3' and (reverse) 5'-CCC CGG GAT TCT GCC TCA CTA CTC-3'; *Myh2* (forward) 5'-AGA ACA TGG AGC AGA CCG TG-3' and (reverse) 5'-TCA TTC CAC AGC ATC GGG AC-3'; *Gapdh* (forward) 5'-AGC AGG CAT CTG AGG GCC CA-3' and (reverse) 5'-TGT TGG GGG CCG AGT TGG GA-3'. Relative gene expression quantification was performed using the ΔΔCt method with *Gapdh* as the reference gene.

## Tissue histology

OCT-embedded frozen muscle was cross-sectioned into 10 µm thick sections, stained with Hematoxylin & Eosin (H&E) or picrosirius red as previously described (*Hammers et al., 2020*), and visualized

with a Leica DMR bright-field microscope equipped with a digital camera (Leica No. DFC480). Image analysis was performed using FIJI software.

## Statistical analysis

Quantified data of this study are displayed as box-and-whisker plots (depicting second and third quartiles with minimum and maximum values) or as histograms of full population distribution, and were analyzed using two-tailed Welch's t-test (α = 0.05; effect size reported as Cohen's d) or one-way ANOVA (effect size reported as $\eta^2$) followed by Tukey post hoc tests (α = 0.05). Power analyses (power = 0.8; α = 0.05) using previous or preliminary data for each measure dictated all sample sizes utilized in this study. No data points were excluded from data analysis during the course of this study.

## Acknowledgements

This work was funded by R01-AR075637 from the National Institute of Arthritis and Musculoskeletal and Skin Diseases (NIAMS), a Wellstone Muscular Dystrophy Cooperative Center grant (U54-AR-052646) from the National Institutes of Health, and Fondation Leducq funding (13CVD04) to HLS. DWH was supported by the Muscular Dystrophy Association (MDA549004) during the course of this work. REC was supported by a National Institutes of Health grant (NIGMS R01 GM134531), EGH was supported by NCI T32CA009156 to the Lineberger Comprehensive Cancer Center, and JAH was supported by the intramural program of the National Heart, Lung, and Blood Institute. The authors thank Dr Melissa Merscham-Banda, Radhika Bhake, Lillian Wright, and Dr Laurence Prunetti for technical assistance during this work.

## Additional information

### Funding

| Funder | Grant reference number | Author |
| --- | --- | --- |
| National Institute of Arthritis and Musculoskeletal and Skin Diseases | R01-AR075637 | H Lee Sweeney |
| National Institute of Arthritis and Musculoskeletal and Skin Diseases | U54-AR-052646 | H Lee Sweeney |
| Fondation Leducq | 13CVD04 | H Lee Sweeney |
| Muscular Dystrophy Association | MDA549004 | David W Hammers |
| National Institute of General Medical Sciences | R01-GM134531 | Richard E Cheney |
| National Heart, Lung, and Blood Institute | | John A Hammer |
| National Cancer Institute | T32CA009156 | Ernest G Heimsath |

The funders had no role in study design, data collection and interpretation, or the decision to submit the work for publication.

### Author contributions

David W Hammers, Conceptualization, Data curation, Formal analysis, Funding acquisition, Investigation, Methodology, Writing – original draft, Writing – review and editing; Cora C Hart, Investigation; Michael K Matheny, Investigation, Methodology, Writing – review and editing; Ernest G Heimsath, Young il Lee, Investigation, Writing – review and editing; John A Hammer, Conceptualization, Methodology, Resources, Writing – review and editing; Richard E Cheney, Conceptualization, Resources,

Writing – review and editing; H Lee Sweeney, Conceptualization, Formal analysis, Funding acquisition, Writing – original draft, Writing – review and editing

### Author ORCIDs
David W Hammers http://orcid.org/0000-0003-2129-4047
John A Hammer III, http://orcid.org/0000-0002-2496-5179
Richard E Cheney http://orcid.org/0000-0001-6565-7888
H Lee Sweeney http://orcid.org/0000-0002-6290-8853

### Ethics

This study was performed in strict accordance with the recommendations in the Guide for the Care and Use of Laboratory Animals of the National Institutes of Health. All of the animals were handled according to approved institutional animal care and use committee (IACUC) protocols of the University of Florida. Protocol #201910602.

### Decision letter and Author response

Decision letter https://doi.org/10.7554/eLife.72419.sa1
Author response https://doi.org/10.7554/eLife.72419.sa2

---

## Additional files

### Supplementary files
• Transparent reporting form

### Data availability

All data generated or analysed during this study are included in the manuscript and supporting file. Source data files have been provided.

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
