## [Decision Letter]

**Acceptance summary:**

This study demonstrates that actin-rich filopodia, generated by class X myosin (Myo10), contribute to myoblast fusion in vitro. Moreover, the authors provide evidence that depletion of Myo10 results in impaired muscle regeneration in vivo. This study provides important new information on the mechanisms of myoblast fusion, and will be of interest to researchers studying cell-cell fusion and muscle biology.

**Decision letter after peer review:**

[Editors’ note: the authors submitted for reconsideration following the decision after peer review. What follows is the decision letter after the first round of review.]

Thank you for submitting your work entitled "Filopodia powered by class X myosin promote fusion of mammalian myoblasts" for consideration by *eLife*. Your article has been reviewed by 3 peer reviewers, including Pekka Lappalainen as the Reviewing Editor and Reviewer #1, and the evaluation has been overseen by a Reviewing Editor and a Senior Editor. The following individual involved in review of your submission has agreed to reveal their identity: Leonid Chernomordik (Reviewer #3).

Our decision has been reached after consultation between the reviewers. Based on these discussions and the individual reviews below, we regret to inform you that your work will not be considered further for publication in *eLife*.

Filopodia have been implicated in cell-cell fusion during muscle development in both flies and zebrafish, but the precise role of these actin-rich protrusions in muscle development remains unknown. Myo10 is an unconventional myosin that is linked to filopodia formation in many cell-types. The study by Hammers et al. reveals that Myo10 is upregulated during myoblast differentiation and muscle regeneration. By taking advantage of both a mouse C2C12 cell-line and conditional Myo10 knockout mice, they demonstrate that Myo10-dependent filopodia are important for myoblast fusion during muscle differentiation in vitro, and provide strong evidence that Myo10 is also critical for muscle development and regeneration in vivo

All three reviewers found this study interesting, and felt that the knockout and knockdown experiments presented in the manuscript convincingly demonstrate the role of Myo10 in myoblast fusion. While reviewer #1 was more positive, the two other reviewers (although still very supportive about the importance of this study) stated the underlying mechanism remains elusive, and thus the manuscript is not suitable for publication in *eLife* in its present form. Therefore, it would be important to provide some mechanistic insight into how Myo10 -generated filopodia contribute to myoblast fusion (by 1. Performing careful, quantitative imaging of the myoblast fusion process, 2. Uncovering if the requirement of Myo10-dependent filopodia in this process is unidirectional or bidirectional, and 3. Characterizing a FERM deletion mutant of Myo10 to elucidate if interaction of this protein with cadherins/integrins is needed for myoblast fusion. ). Moreover, some imaging experiments, including the ones on myomaker and myomixer, are not particularly convincing, and whether these proteins are indeed enriched in filopodia, remains unclear. Thus, the imaging data should be improved throughout the manuscript, and one should provide careful quantification of the data to support the conclusions. Detailed comments by the three reviewers are below.

Because the policy of *eLife* is to invite a revision only if the suggested experiments can be carried out in 2-3 months, we cannot unfortunately offer to publish this study in *eLife*. However, all three reviewers found this work very interesting and important. Thus, if you can address main points listed by the three reviewers, and provide some mechanistic insight into how Myo10/filopodia contribute to myoblast fusion, we will be glad to consider a new submission on this topic for publication in *eLife*. In such case, the new submission would be evaluated by the three original reviewers.

*Reviewer #1:*

Actin-rich filopodia-like protrusions participate in myoblast fusion, and at least in *Drosophila* filopodial proteins Enabled and IRSp53 are critical for myoblast-myotube fusion. However, the precise molecular composition of filopodia that contribute to myoblast fusion, especially in vertebrates, is incompletely understood. Here, Hammers et al. provide evidence that class X myosin (Myo10) is an important component of myoblast fusion in mammals. They show that expression of Myo10 is upregulated during myoblast differentiation and muscle regeneration, and that Myo10 drives the formation of filopodia in myoblasts. Importantly, they demonstrate that deletion of Myo10 leads to defects in myoblast fusion in vitro and for muscle regeneration in vivo. This manuscript provides important new information on the mechanisms of myoblast fusion. However, there are few points that should be addressed to strengthen the study.

The images on the localization of Myomaker and Myomixer along filopodia in differentiating muscle cells are not convincing. For example, from Figure 5B it appears that Myomixer does not localize to filopodia and also Myomaker vesicles display only very occasional localization to filopodia. Thus, the authors should repeat these experiments to obtain better quality images, or tone down their conclusions about localization of myogenic fusion proteins to filopodia.

*Reviewer #2:*

Filopodia have been strongly implicated in mediating cell-cell fusion during muscle development in both flies and zebrafish. The exact role of these cellular protrusions in muscle development remains unknown. Hammers et al. implicate filopodia generated by the unconventional myosin Myo10 as key mediators of myoblast fusion during muscle development and suggest that they are important for delivering the muscle fusogens Myomaker and Myomixer to target cells. Most interestingly, the work also strongly implicates filopodia in having a major role during muscle regeneration following injury.

The authors take good advantage of both the mouse C2C12 line and a conditional Myo10 KO mouse to highlight the importance of filopodia in muscle. The results clearly and convincingly show that Myo10-dependent filopodia play a critical role in myoblast fusion during muscle differentiation in vitro and also strongly implicate them in fusion in vivo, providing a new perspective on the mechanism of muscle development and regeneration. However, while the findings are highly interesting and the results supportive of the author's conclusions, the paper itself is lacking compelling data showing that filopodia-driven contact between myoblasts is the key initiating event or plays a significant role in promoting fusion. This conclusion is logically consistent with all of the data and a number of images and videos are quite suggestive. However, few examples of filopodia interacting with a target cell are shown, they are not always clear and, most importantly, there is a lack of quantitative support for a high frequency of 'first contact' or significant contact between filopodia and their target cell. Furthermore, the broad membrane localization of Myomaker and Myomixer to the surface of differentiating myoblasts makes it difficult to know if the presence of these fusogens in filopodia is really critical for fusion (i.e. are filopodia playing a major role in delivering these to the target cell for fusion). The available data, both from this work and that of others, suggests that there is a high likelihood that filopodia are key critical drivers of myoblast fusion. The inclusion of stronger data showing their interaction with a target cell would make this conclusion more compelling.

1. Thin extensions the size of filopodia are present at the front, sides and rear of undifferentiated, migrating myoblasts, as visualized in cells expressing RFP- or GFP-CAXX (Figure 1A; Figure S1 A). The filopodia are said to be preferentially extended from the leading edge of the cell and the observed lateral and rear extensions are said to be retraction fibers. While this seems reasonable and the single video provided is consistent with this, it is difficult to assess as no information is provided about any quantitative characterization to support this conclusion.

Details about the method for measuring projections lengths (Figure S1B, Figure 3E) are also missing.

2. Video 8 shows fusion of the lateral edges of myotubes. The initial projection is clear to see, but the video is rather choppy and it is difficult to clearly follow the interaction between filopodia or membrane extensions of the two cells. Similarly, it is hard to appreciate the membrane fusion events in Video 9.

3. Integrin b1 staining is indeed enriched at the front of the differentiating myoblast, but it is hard to visualize integrin B1 within the Myo10-positive filopodia themselves (Figure 2D).

4. A striking loss of myoblast fusion is observed after 7 days of differentiation when Myo10 is depleted (Figure 3F). Apart from the loss of filopodia, which is also significant (Figure 3E), are there any other noticeable defects in the cells, such as reduced adhesion that has been reported by others such as in the Zhang et al. paper? Is there any impact on the migration of the knock-down cells? Could either of these potential phenotypes, if present, account for or contribute to the reduced levels of fusion?

5. The text refers to Myo10 containing filopodia with Myomixer puncta in regenerating Pax-WT muscle and that these while these projections are missing from Pax-Myo10 cKO muscle (lines 264-266), referring to Figure S5F, G. Those panels are missing from the supplemental figure.

6. It is notable that no gross abnormalities of the musculature have been reported for the Myo10 KO mouse. If filopodia are indeed critical for muscle fusion and, by extension, development then wouldn't one expect to see an obvious muscle defect in these mutants?

*Reviewer #3:*

1) The authors may consider addressing some of the questions related to the mechanistic role of Myo10 and Myo10 filopodia in fusion. Just as examples, can Myo10 contribute to fusion via its involvement in cell adhesion? Through its FERM domain, Myo10 interacts with N-cadherin (Lai et al., 2015, Front Cell Neurosci. 9: 326), a protein implicated in myoblast fusion (Knudsen et al., 1990, Experim. Cell Res, 188, 175-184). FERM domain is also responsible for Myo10 interactions with β-integrin at filopodia tips (Alieva et al., 2019, Nature Commun. 10, Article number: 3593). Β-integrins have been also implicated into myoblast fusion (Schwander et al., 2003 Dev Cell 4(5):673-85). Will filopodia induced by truncated Myo10 lacking FERM domain support fusion?

Are Myo10 and Myo10-filopedia required by both fusing cells, or by only one of the cells?

Can Myo10-generated filopodia promote fusion by concentrating fusogenic proteins? It would be interesting to compare the amounts of Myomaker and Myomerger associated with Myo10-generated filopodia vs. those at the plasma membrane along the cell body and at lamellipodial extensions formed by the cells lacking Myo10? Can Myo10 deficiency be rescued by overexpressing "fusogenic" proteins?

Again, these are just examples of questions that may deepen understanding of the role of Myo10 in myoblast fusion.

2) In Figure 1, the authors compare undifferentiated (and mononucleated) C2C12 cells and multinucleated myotubes. Since multinucleated cells are obviously quite different from mononucleated cells, it would be interesting and perhaps more informative, to compare undifferentiated C2C12 cells with yet mononucleated differentiated C2C12 cells.

3) Figure 1F shows myoblast utilizing a lamellipodial extension with protrusions to fuse with an adjacent cell. How many fusion events are caught early enough to resolve cell-cell bridging projections and is there any correlation between location and morphology of the projections (dorsal vs. lamellipodial extensions) and the probability of their association with fusion site?

4) Figure S4. According to Millay et al., 2013, myomaker gene in C2C12 cells just starts to be expressed after 1 day of differentiation. Do you expect it to already be well expressed at the surface of C2C12 cells? The specificity of immunostaining with Myomaker antibodies in your experiments should be verified by controls with myomaker-deficient myoblasts or, at least, with proliferating C2C12 cells.

5) Figure 5B. The legend describes the images as: "Myomaker and Myomixer puncta are found localized to Myo10-filled filopodia of differentiating myoblasts." While the images show that Myomaker and Myomixer are present in filopodia, I see no indication in these images that Myomaker and Myomixer are enriched in the filopodia. It is difficult to say without analysis but filopodia seem to show less staining for these proteins than the body of the cell.

[Editors’ note: further revisions were suggested prior to acceptance, as described below.]

Thank you for submitting your article "Filopodia powered by class X myosin promote fusion of mammalian myoblasts" for consideration by *eLife*. Your article has been reviewed by 3 peer reviewers, including Pekka Lappalainen as the Reviewing Editor and Reviewer #1, and the evaluation has been overseen by a Reviewing Editor and Anna Akhmanova as the Senior Editor. The following individual involved in review of your submission has agreed to reveal their identity: Leonid Chernomordik (Reviewer #3).

Essential revisions:

*Reviewer #1:*

The authors have satisfactorily addressed my previous concerns. The revised manuscript is of good technical quality, and provides important new information on the mechanism of myoblast fusion.

*Reviewer #2:*

The authors have satisfactorily responded to the comments raised in the initial review and the now include data showing that a cargo binding mutant of Myo10 cannot support fusion strengthens the work. These new results strongly support a role for filopodia in promoting myoblast fusion, especially in the context of injury, and reveal a role for an as yet unknown Myo10 cargo in mediating that process.

There are some remaining issues that need to be addressed where possible.

1) A mutant Myo10 lacking the cargo binding domain rescues filopodia formation but not myoblast fusion (line 208; Figure S3). How efficient is the rescue of filopodia, is it comparable to the full-length myosin? The one image shown suggests that the cells expressing the mutant may make an excess of filopodia but that may just be an extreme example. If one quantifies filopodia formation per cell is the rescue comparable to the full-length myosin? Has expression of the truncated Myo10 been validated by western blotting?

2) The authors discuss their observation of 'projections' in differentiating myotubes and initially state that these are filopodia (bottom of pg 5). They then go on to say that projections induce fusion. It is not clear in this section of the paper if the authors consider all projections to be filopodia and what the difference is between a projection and protrusion. The text (and legend for Figure 1) should clearly describe what type of extension is being discussed/examined (this should be done throughout the manuscript).

3) The time of differentiation is not always clearly stated in the text or indicated on figures or in the legend (e.g. Figure 3, S2, S3, S4; Figure 5B). The authors should specifically state the differentiation time point in text and legends (when not shown on figure panels) to make it more transparent for readers.

4) Legend, Figure S2

– line 895 – the legend refers to the Myo10 promoter depicted in panel B, this should be corrected to "D" where the promoter illustration is shown.

– lines 893-896 more information should be provided about the images shown in panel G

5) Quantification of filopodia and measurements of lengths is done using ImageJ (lines 520 – 522). Are these measurements done on images taken from a single plane, if so then which one, or from projections of a Z-stack? This should be indicated in the Methods.

6) The information for the Myo10 antibody used in this study (line 537) should be verified – the source is not provided and the RRID number does not correspond to a Myo10 antibody in the database.

*Reviewer #3:*

The revision has strengthened this interesting and important study and addressed my comments/ questions. I am very impressed by new experiments added in the revised MS (Figure S4) convincingly showing that suppressing filopodia formation in half of the myoblasts dramatically inhibits fusion. We still do not know why myoblast fusion depends on filopodia formation and it has been suggested that filopodia promote fusion merely by allowing closer filopodium-plasma membrane apposition and/or by generating at the top of filopodium regions of a strong local bending of membrane lipid bilayers (Chen, E.H., Invasive Podosomes and Myoblast Fusion, Curr Top Membr. 2011; 68: 235-258). Finding that suppressing filopodia formation in half of myoblasts inhibits fusion much more than 2 times argues against these mechanisms. Moreover, finding that like myomaker expression, Myo10 (and, by extension, the ability to form filopodia) is required bilaterally for myoblast fusion suggests that myomaker function depends on filopodium formation. The authors may want to emphasize these intriguing findings and comment on them in the Discussion.

---

## [Author Response]

[Editors’ note: the authors resubmitted a revised version of the paper for consideration. What follows is the authors’ response to the first round of review.]

Reviewer #1:Actin-rich filopodia-like protrusions participate in myoblast fusion, and at least in *Drosophila* filopodial proteins Enabled and IRSp53 are critical for myoblast-myotube fusion. However, the precise molecular composition of filopodia that contribute to myoblast fusion, especially in vertebrates, is incompletely understood. Here, Hammers et al. provide evidence that class X myosin (Myo10) is an important component of myoblast fusion in mammals. They show that expression of Myo10 is upregulated during myoblast differentiation and muscle regeneration, and that Myo10 drives the formation of filopodia in myoblasts. Importantly, they demonstrate that deletion of Myo10 leads to defects in myoblast fusion in vitro and for muscle regeneration in vivo. This manuscript provides important new information on the mechanisms of myoblast fusion. However, there are few points that should be addressed to strengthen the study.The images on the localization of Myomaker and Myomixer along filopodia in differentiating muscle cells are not convincing. For example, from Figure 5B it appears that Myomixer does not localize to filopodia and also Myomaker vesicles display only very occasional localization to filopodia. Thus, the authors should repeat these experiments to obtain better quality images, or tone down their conclusions about localization of myogenic fusion proteins to filopodia.

Thank you for noting this concern. We agree that there is a challenge in interpreting whether these highly-expressed proteins are being actively associated with filopodia or freely diffusible across the membrane that encloses a filopodium. In the current manuscript, we address this issue by investigating where these proteins localize in the insoluble, actin-associated cellular fraction of differentiated myotubes. As you will see in Figure 5 panels C and D, both Myomixer and Myomaker puncta can be found co-localizing with Myo10 on actin bundles representing the remnants of myotube filopodia. While we have also demonstrated the specificity of the commercially-available Myomaker and Myomixer antibodies used for this assessment using undifferentiated myoblasts transfected with plasmids encoding each of the proteins (Figure S7A-B), this experiment was also repeated using Flag-tagged versions of Myomaker and Myomixer, that have been previously demonstrated to retain protein viability, in case there are remaining concerns about the specificity of the antibodies used. As shown in Figure S8, these Flag-tagged Myomaker and Myomixer constructs demonstrate similar localization patterns as the native protein immunofluorescence in both complete cells and the insoluble cellular fraction. We have, therefore, interpreted these data to confirm that Myomaker and Myomixer are associated with the cytoskeletal aspects of muscle filopodia.

Reviewer #2:Filopodia have been strongly implicated in mediating cell-cell fusion during muscle development in both flies and zebrafish. The exact role of these cellular protrusions in muscle development remains unknown. Hammers et al. implicate filopodia generated by the unconventional myosin Myo10 as key mediators of myoblast fusion during muscle development and suggest that they are important for delivering the muscle fusogens Myomaker and Myomixer to target cells. Most interestingly, the work also strongly implicates filopodia in having a major role during muscle regeneration following injury.The authors take good advantage of both the mouse C2C12 line and a conditional Myo10 KO mouse to highlight the importance of filopodia in muscle. The results clearly and convincingly show that Myo10-dependent filopodia play a critical role in myoblast fusion during muscle differentiation in vitro and also strongly implicate them in fusion in vivo, providing a new perspective on the mechanism of muscle development and regeneration. However, while the findings are highly interesting and the results supportive of the author's conclusions, the paper itself is lacking compelling data showing that filopodia-driven contact between myoblasts is the key initiating event or plays a significant role in promoting fusion. This conclusion is logically consistent with all of the data and a number of images and videos are quite suggestive. However, few examples of filopodia interacting with a target cell are shown, they are not always clear and, most importantly, there is a lack of quantitative support for a high frequency of 'first contact' or significant contact between filopodia and their target cell. Furthermore, the broad membrane localization of Myomaker and Myomixer to the surface of differentiating myoblasts makes it difficult to know if the presence of these fusogens in filopodia is really critical for fusion (i.e. are filopodia playing a major role in delivering these to the target cell for fusion). The available data, both from this work and that of others, suggests that there is a high likelihood that filopodia are key critical drivers of myoblast fusion. The inclusion of stronger data showing their interaction with a target cell would make this conclusion more compelling.1. Thin extensions the size of filopodia are present at the front, sides and rear of undifferentiated, migrating myoblasts, as visualized in cells expressing RFP- or GFP-CAXX (Figure 1A; Figure S1 A). The filopodia are said to be preferentially extended from the leading edge of the cell and the observed lateral and rear extensions are said to be retraction fibers. While this seems reasonable and the single video provided is consistent with this, it is difficult to assess as no information is provided about any quantitative characterization to support this conclusion.Details about the method for measuring projections lengths (Figure S1B, Figure 3E) are also missing.

Thank you for this suggestion. Quantitative data of the cellular projections of undifferentiated myoblasts are now found in Figures S1B-C. Additionally, details of how projection lengths were measured are now found Lines 520-522 of the current manuscript.

2. Video 8 shows fusion of the lateral edges of myotubes. The initial projection is clear to see, but the video is rather choppy and it is difficult to clearly follow the interaction between filopodia or membrane extensions of the two cells. Similarly, it is hard to appreciate the membrane fusion events in Video 9.

Due to this concern, these videos have been removed. In their place, you will now find Video S8 that provides a more detailed visualization of cell fusion occurring at the myotube lateral edge.

3. Integrin b1 staining is indeed enriched at the front of the differentiating myoblast, but it is hard to visualize integrin B1 within the Myo10-positive filopodia themselves (Figure 2D).

We agree that the Itgb1 staining provided in the previous submission was hard to see and did not necessarily provide useful information. That sub-image has been removed.

4. A striking loss of myoblast fusion is observed after 7 days of differentiation when Myo10 is depleted (Figure 3F). Apart from the loss of filopodia, which is also significant (Figure 3E), are there any other noticeable defects in the cells, such as reduced adhesion that has been reported by others such as in the Zhang et al. paper? Is there any impact on the migration of the knock-down cells? Could either of these potential phenotypes, if present, account for or contribute to the reduced levels of fusion?

Observations of differentiating Myo10 knockdown cells do not indicate that cellular adhesion or migration behaviors are affected by loss of Myo10 in vitro. Therefore, we do not believe these aspects are responsible for the loss of myoblast fusion observed.

5. The text refers to Myo10 containing filopodia with Myomixer puncta in regenerating Pax-WT muscle and that these while these projections are missing from Pax-Myo10 cKO muscle (lines 264-266), referring to Figure S5F, G. Those panels are missing from the supplemental figure.6. It is notable that no gross abnormalities of the musculature have been reported for the Myo10 KO mouse. If filopodia are indeed critical for muscle fusion and, by extension, development then wouldn't one expect to see an obvious muscle defect in these mutants?

Our apologies for this mis-label. Those images are found in Figure 4E-F. As is now elaborated in more detail in our Discussion section (Lines 370-377), we suspect that filopodia facilitate developmental processes by increasing the probability of proper cellular connections being made. Likely, the incredibly dense cellular content of developing embryonic muscle is able to compensate for loss of Myo10/filopodia during embryonic development, as adequate cell-to-cell contact is likely met to induce myoblast fusion.

Reviewer #3:1) The authors may consider addressing some of the questions related to the mechanistic role of Myo10 and Myo10 filopodia in fusion. Just as examples, can Myo10 contribute to fusion via its involvement in cell adhesion? Through its FERM domain, Myo10 interacts with N-cadherin (Lai et al., 2015, Front Cell Neurosci. 9: 326), a protein implicated in myoblast fusion (Knudsen et al., 1990, Experim. Cell Res, 188, 175-184). FERM domain is also responsible for Myo10 interactions with β-integrin at filopodia tips (Alieva et al., 2019, Nature Commun. 10, Article number: 3593). Β-integrins have been also implicated into myoblast fusion (Schwander et al., 2003 Dev Cell 4(5):673-85). Will filopodia induced by truncated Myo10 lacking FERM domain support fusion?Are Myo10 and Myo10-filopedia required by both fusing cells, or by only one of the cells?Can Myo10-generated filopodia promote fusion by concentrating fusogenic proteins? It would be interesting to compare the amounts of Myomaker and Myomerger associated with Myo10-generated filopodia vs. those at the plasma membrane along the cell body and at lamellipodial extensions formed by the cells lacking Myo10? Can Myo10 deficiency be rescued by overexpressing "fusogenic" proteins?Again, these are just examples of questions that may deepen understanding of the role of Myo10 in myoblast fusion.

Thank you for these insightful suggestions. In response, we have performed two key experiments that provide more insight into the involvement of Myo10 in muscle cell fusion.

First, we have investigated the requirement of Myo10 cargo binding for the promotion of myoblast fusion using a Myo10 construct lacking much of the C-terminal tail, including the PEST, PH, MyTH4, and FERM domains (referred to as Myo10ΔCBD). As shown in Figure S3A, cells expressing this construct do form filopodia, as the Myo10 motor domain and anti-parallel coiled-coil are not affected. Expression of this truncated Myo10, however, does not rescue myoblast fusion, as is seen using the full-length Myo10 construct. These data are currently found in Figure S3B.

Secondly, we have sought to determine whether Myo10 is required by one of the fusing cells or both fusing cells to promote multinucleated muscle formation. As shown in Figure S4A, we distinctly labeled control and Myo10 KD cells with fluorescent proteins and performed a co-culture experiment to determine if chimeric myotubes are formed. To our surprise, evidence of fusion between control and Myo10 KD myoblasts is exceedingly rare (only 1 instance found; see Figure S4B). These data (found in Figure S4C) suggest that is it highly favorable for Myo10 to be present on both cells of a muscle fusion event.

2) In Figure 1, the authors compare undifferentiated (and mononucleated) C2C12 cells and multinucleated myotubes. Since multinucleated cells are obviously quite different from mononucleated cells, it would be interesting and perhaps more informative, to compare undifferentiated C2C12 cells with yet mononucleated differentiated C2C12 cells.

Once myoblasts undergo differentiation, begin expressing Myo10, and exhibit cellular elongation into a myotube (as shown in Video S11), the cellular projection are very similar independently of whether the myotube has one or several nuclei. Notable differences in cellular projection numbers and sizes are only found between undifferentiated and differentiated cells within the conditions studied in this manuscript.

3) Figure 1F shows myoblast utilizing a lamellipodial extension with protrusions to fuse with an adjacent cell. How many fusion events are caught early enough to resolve cell-cell bridging projections and is there any correlation between location and morphology of the projections (dorsal vs. lamellipodial extensions) and the probability of their association with fusion site?

Unfortunately, visualizing active cell fusion events in fluorescence-based live cell experiments is a low-probability occurrence that is sensitive to the frequency of image acquisition, even in low laser-power conditions. Therefore, catching the precise structures leading to cellular fusion is incredibly difficult. In the current submission, we have included a new sequence into Figure 1F and Video S8 that depicts a cellular fusion event initiated by a myotube lateral edge that clearly exhibits several filopodia that engage with the target cell prior to fusion. From the limited visual information we have on the phenomenon, it appears the probability of fusion is equal from lateral edges, dorsal/belly surfaces, and lamellipodial extension of myotubes, and really depends on where the fusion target cell is located in relation to the fusion-initiating myotube.

4) Figure S4. According to Millay et al., 2013, myomaker gene in C2C12 cells just starts to be expressed after 1 day of differentiation. Do you expect it to already be well expressed at the surface of C2C12 cells? The specificity of immunostaining with Myomaker antibodies in your experiments should be verified by controls with myomaker-deficient myoblasts or, at least, with proliferating C2C12 cells.

We suspect Myomaker can be externalized to the plasma membrane (presumably from Golgi-derived vesicles, as demonstrated by Gamage et al.) soon after expression begins, as Myomixer is confirmed to be externalized early in differentiation, using antibodies with C-terminal epitopes (Figures S7-8).

In response to your concern, we have included our antibody testing images using undifferentiated myoblasts transfected with either Myomaker or Myomixer. These data suggest that the antibodies have specificity within the concentration/signal range used for this study. To further confirm that our observations obtained with Myomaker and Myomixer antibodies are not artifacts of non-specfic antibodies, the studies were repeated using Flag-tagged versions of the proteins that have been previously reported to retain protein function (Figure S8).

5) Figure 5B. The legend describes the images as: "Myomaker and Myomixer puncta are found localized to Myo10-filled filopodia of differentiating myoblasts." While the images show that Myomaker and Myomixer are present in filopodia, I see no indication in these images that Myomaker and Myomixer are enriched in the filopodia. It is difficult to say without analysis but filopodia seem to show less staining for these proteins than the body of the cell.

As mentioned in our response to Reviewer #1, we have attempted to address this concern by investigating the localization of these proteins in the insoluble cellular fraction to determine if they can be evidenced to be associated with the cytoskeleton and/or Myo10 of filopodia. As is now included in Figure 5C-D, native Myomixer and Myomaker can be found co-localizing with Myo10 on actin bundles. The Flag-tagged version of the two proteins show the same localization pattern, as well (Figure S8B-C). We believe these data provide evidence that there is an association between muscle filopodia and these fusion proteins.

[Editors’ note: what follows is the authors’ response to the second round of review.]

Reviewer #2:The authors have satisfactorily responded to the comments raised in the initial review and the now include data showing that a cargo binding mutant of Myo10 cannot support fusion strengthens the work. These new results strongly support a role for filopodia in promoting myoblast fusion, especially in the context of injury, and reveal a role for an as yet unknown Myo10 cargo in mediating that process.There are some remaining issues that need to be addressed where possible.1) A mutant Myo10 lacking the cargo binding domain rescues filopodia formation but not myoblast fusion (line 208; Figure S3). How efficient is the rescue of filopodia, is it comparable to the full-length myosin? The one image shown suggests that the cells expressing the mutant may make an excess of filopodia but that may just be an extreme example. If one quantifies filopodia formation per cell is the rescue comparable to the full-length myosin? Has expression of the truncated Myo10 been validated by western blotting?

In response to this inquiry, we conducted a quantification of filopodia resulting from expression of these Myo10 constructs. These data showing filopodia numbers and filopodia lengths are now found in Figure S3C and D, respectively. In summary, both Myo10 constructs promote similar numbers of filopodia, while the truncated version results in slightly shorter filopodia lengths. Accordingly, a more representative image has been included in the image panel for RFP-Myo10. Also, we have provided an immunoblot verifying that these constructs express protein products of the appropriate size, found in Figure S3B.

2) The authors discuss their observation of 'projections' in differentiating myotubes and initially state that these are filopodia (bottom of pg 5). They then go on to say that projections induce fusion. It is not clear in this section of the paper if the authors consider all projections to be filopodia and what the difference is between a projection and protrusion. The text (and legend for Figure 1) should clearly describe what type of extension is being discussed/examined (this should be done throughout the manuscript).

During the drafting of this manuscript, we carefully considered how we would distinguish filopodia, a very specific cellular structure, from non-specific cellular structures (such as retraction fibers), which we refer to as either projections or protrusions (interchangeable, non-specific terms). We decided the most appropriate definition of filopodia is an actively elongating thin cellular projection that is dependent on Myo10. Based on this strict definition, we could not confirm that these structures were indeed filopodia until the Myo10 loss of function experiment was conducted (Lines 189-191). Accordingly, we refer to the entire population of these structures as projections/protrusions and verified Myo10-driven structures as filopodia throughout the manuscript to emphasize this distinction.

3) The time of differentiation is not always clearly stated in the text or indicated on figures or in the legend (e.g. Figure 3, S2, S3, S4; Figure 5B). The authors should specifically state the differentiation time point in text and legends (when not shown on figure panels) to make it more transparent for readers.

Thank you for noting these omissions. They have been added to the current version of the manuscript.

4) Legend, Figure S2– line 895 – the legend refers to the Myo10 promoter depicted in panel B, this should be corrected to "D" where the promoter illustration is shown.– lines 893-896 more information should be provided about the images shown in panel G

Thank you also for noting these errors. They, too, have been corrected.

5) Quantification of filopodia and measurements of lengths is done using ImageJ (lines 520 – 522). Are these measurements done on images taken from a single plane, if so then which one, or from projections of a Z-stack? This should be indicated in the Methods.

The images were quantified as Z-stacks. This detail has now been added to the manuscript.

6) The information for the Myo10 antibody used in this study (line 537) should be verified – the source is not provided and the RRID number does not correspond to a Myo10 antibody in the database.

We apologize for this error. The information of the Myo10 antibody used for immunoblotting has been clarified.

Reviewer #3:The revision has strengthened this interesting and important study and addressed my comments/ questions. I am very impressed by new experiments added in the revised MS (Figure S4) convincingly showing that suppressing filopodia formation in half of the myoblasts dramatically inhibits fusion. We still do not know why myoblast fusion depends on filopodia formation and it has been suggested that filopodia promote fusion merely by allowing closer filopodium-plasma membrane apposition and/or by generating at the top of filopodium regions of a strong local bending of membrane lipid bilayers (Chen, E.H., Invasive Podosomes and Myoblast Fusion, Curr Top Membr. 2011; 68: 235-258). Finding that suppressing filopodia formation in half of myoblasts inhibits fusion much more than 2 times argues against these mechanisms. Moreover, finding that like myomaker expression, Myo10 (and, by extension, the ability to form filopodia) is required bilaterally for myoblast fusion suggests that myomaker function depends on filopodium formation. The authors may want to emphasize these intriguing findings and comment on them in the Discussion.

Thank you for this helpful discussion. At your suggestion, we have added a new sentence (lines 396-399) to the Discussion section pointing out the possibility that Myomaker’s function may depend on Myo10’s role filopodia formation and cargo binding ability.